# Germline VRC01 antibody recognition of a modified clade C HIV-1 envelope trimer and a glycosylated HIV-1 gp120 core

Andrew J Borst[1], Connor E Weidle[2], Matthew D Gray[2], Brandon Frenz[1], Joost Snijder[1], M Gordon Joyce[3], Ivelin S Georgiev[3], Guillaume BE Stewart-Jones[3], Peter D Kwong[3], Andrew T McGuire[2], Frank DiMaio[1], Leonidas Stamatatos[2,4]*, Marie Pancera[2,3]*, David Veesler[1]*

[1]Department of Biochemistry, University of Washington, Seattle, United States; [2]Vaccine and Infectious Disease Division, Fred Hutchinson Cancer Research Center, Seattle, United States; [3]Vaccine Research Center, National Institute of Allergy and Infectious Diseases, National Institutes of Health, Bethesda, United States; [4]Department of Global Health, University of Washington, Seattle, United States

*For correspondence:
lstamata@fredhutch.org (LS);
mpancera@fredhutch.org (MP);
dveesler@uw.edu (DV)

Competing interests: The authors declare that no competing interests exist.

**Abstract** VRC01 broadly neutralizing antibodies (bnAbs) target the CD4-binding site (CD4$_{BS}$) of the human immunodeficiency virus-1 (HIV-1) envelope glycoprotein (Env). Unlike mature antibodies, corresponding VRC01 germline precursors poorly bind to Env. Immunogen design has mostly relied on glycan removal from trimeric Env constructs and has had limited success in eliciting mature VRC01 bnAbs. To better understand elicitation of such bnAbs, we characterized the inferred germline precursor of VRC01 in complex with a modified trimeric 426c Env by cryo-electron microscopy and a 426c gp120 core by X-ray crystallography, biolayer interferometry, immunoprecipitation, and glycoproteomics. Our results show VRC01 germline antibodies interacted with a wild-type 426c core lacking variable loops 1–3 in the presence and absence of a glycan at position Asn276, with the latter form binding with higher affinity than the former. Interactions in the presence of an Asn276 oligosaccharide could be enhanced upon carbohydrate shortening, which should be considered for immunogen design.

DOI: https://doi.org/10.7554/eLife.37688.001

## Introduction

Despite the tremendous impact of HIV-1 on human health, no efficacious HIV-1 vaccine currently exists. The HIV-1 envelope (Env) glycoprotein is a class-I fusion protein responsible for host attachment and fusion of the viral and cellular membranes (*Dalgleish et al., 1984*). Following expression, Env trimerizes and undergoes furin-mediated cleavage to yield non-covalent gp120-gp41 pre-fusion trimers anchored in the viral membrane (*Haim et al., 2013*). As the sole target of neutralizing antibodies, Env is the focus of intense interest for current vaccine design initiatives. However, HIV-1 Env relies on multiple mechanisms of immune evasion – including dense glycosylation, sequence variation, conformational masking, and presentation of decoy epitopes (*Burton and Mascola, 2015*; *Cuevas et al., 2015*; *Jardine et al., 2015*; *Kwong et al., 2002*; *Wei et al., 2003*; *Zhou et al., 2017*). For these reasons, development of an Env-based vaccine capable of eliciting broadly neutralizing antibodies (bnAbs) has proven challenging.

The VRC01-class of bnAbs is of particular interest for HIV-1 vaccine development due to the exceptional potency and breadth of several of its well-characterized members (*Huang et al., 2016*; *Zhou et al., 2015*). These bnAbs derive from the VH1-2 variable heavy chain gene (*Scheid et al., 2011*; *Wu et al., 2011*), have been isolated from multiple HIV-1-infected patients (*Zhou et al.,*

*2013*), and putative non-mutated precursors have been identified in naïve individuals (*Jardine et al., 2016a*). VRC01-class bnAbs are characterized by an unusually short five amino-acid light chain complementary-determining region (CDR) L3 loop (*Zhou et al., 2015*) and much higher levels of somatic hyper-mutation than antibodies targeting other pathogens (*Wu et al., 2015*). They bind the CD4-binding site (CD4$_{BS}$) in a way reminiscent of the interactions formed with the viral receptor CD4, making extensive CDRH2-mediated contacts while also exhibiting multiple amino acid alterations in the CDRL1 loop relative to germline precursors (*Wu et al., 2015*; *Zhou et al., 2013*). Although *N*-linked glycosylation sites (NLGSs) that surround the CD4$_{BS}$ sterically limit recognition by bnAbs (*Zhou et al., 2017*), particularly those present at position Asn276 in Loop D and along the V5 loop, mature VRC01 bnAbs overcome this barrier and potently neutralize numerous HIV-1 viral clades (*Zhou et al., 2017*; *Huang et al., 2016*; *Stewart-Jones et al., 2016*; *Wu et al., 2015*). In contrast, the inferred germline precursors of VRC01-class bnAbs lack detectable binding to trimeric Env constructs harboring glycans at these locations (*Jardine et al., 2013*; *McGuire et al., 2016*; *McGuire et al., 2013*; *Medina-Ramírez et al., 2017b*; *Stamatatos et al., 2017*).

Whereas most recombinant trimeric Env antigens do not bind germline precursors of VRC01-class bnAbs, a few recently designed constructs have been shown to bind and activate this specific class of B cell receptors (BCRs) (*Jardine et al., 2013*; *McGuire et al., 2016*; *McGuire et al., 2013*; *Medina-Ramírez et al., 2017a*). We previously engineered a trimeric HIV-1 Env protein able to bind most VRC01-class precursors (*McGuire et al., 2013*). This construct was a trimeric gp140 protein derived from the clade C 426c virus and lacked variable loops 1, 2, and 3, along with the putative NLGSs at positions Asn276 (loop D), Asn460, and Asn463 (V5 loop) (*McGuire et al., 2016*). Other constructs have also been engineered to engage the inferred precursors of VRC01-class bnAbs, all of which harbored mutations eliminating the NLGSs in loop D (at position Asn276) and in the V5 loop (*Briney et al., 2016*; *Jardine et al., 2013*; *McGuire et al., 2016*; *McGuire et al., 2013*; *Medina-Ramírez et al., 2017a*; *Tian et al., 2016*). Additionally, a gp120 core derived from the 01dG5 clade virus, which naturally lacks a glycan at position Asn276, was also shown to engage the inferred germline precursor of the VRC01 antibody (VRC01$_{GL}$) (*Wu et al., 2015*). Although such glycan-depleted 'germline-targeting' immunogens activate B cells expressing germline VRC01-class BCRs in vivo (*Briney et al., 2016*; *Dosenovic et al., 2015*; *Tian et al., 2016*), they largely fail to elicit mature antibodies capable of bypassing the restrictions imposed by the glycan at position Asn276 (*Zhou et al., 2017*). However, a recent study demonstrated the successful elicitation of CD4$_{BS}$-targeted antibodies, distinct from the VRC01 lineage, upon immunization of rabbits with an engineered clade C Env trimer (*Dubrovskaya et al., 2017*).

To better understand the potential avenues of elicitation of VRC01-class bnAbs, we structurally characterized complexes between VRC01$_{GL}$ and two clade C Env constructs using a combination of cryo-electron microscopy (cryoEM) and X-ray crystallography. One of the constructs is a soluble trimeric 426c SOSIP with three NLGSs removed at positions Asn276, Asn460, and Asn463, and is based on our prior work (*McGuire et al., 2016*; *McGuire et al., 2013*). The second construct is a monomeric 426c core containing all wild-type NLGSs (including those at positions Asn276, Asn460, and Asn463), but lacks variable loops 1, 2, and 3. The 426c strain naturally lacks NLGSs surrounding the CD4$_{BS}$ at positions Asn234 and Asn362(363), which are present in other clades. Our structural analysis revealed that the absence of these glycans leads to a reduction of local oligosaccharide density in the vicinity of the NLGS at position Asn276. Integrating this data with biolayer interferometry (BLI) assays and glycoproteomics, we demonstrate here that VRC01$_{GL}$ could bind to a 426c core construct in the presence of all naturally occurring NLGSs surrounding the CD4$_{BS,}$ including the NLGS at position Asn276 and with its associated glycan. We also show the affinity of VRC01$_{GL}$ for the 426c core could be modulated by altering protein expression conditions to enrich for longer glycans, and also by shortening glycans via endoglycosidase treatment. These results suggest that priming of VRC01-class bnAbs may be possible using an HIV-1 gp120 derivative containing a glycan at position Asn276. Consequently, future epitope-based vaccine design strategies utilizing a 426c core preserving all NLGSs may be a promising route for guiding elicitation of VRC01-class bnAbs.

## Results

### CryoEM structure of VRC01$_{GL}$ in complex with a modified 426c HIV-1 SOSIP glycoprotein trimer

Based on the known enhanced ability of VRC01$_{GL}$ (and related germline antibodies) to bind 426c constructs lacking putative NLGSs at positions Asn276, Asn460, and Asn463 (*McGuire et al., 2013*), and the lack of detectable binding to 426c DS-SOSIP (*Figure 1A*), we engineered a modified 426c DS-SOSIP trimer recapitulating the aforementioned glycan depletion mutations for structural analysis. This construct harbors the S278A, T462A and T465A mutations, abolishing the corresponding NLGSs and enabling binding to VRC01$_{GL}$ (*Figure 1B*, *Figure 1—figure supplement 1*). It also contains the SOSIP (*Sanders et al., 2002*) and the 201C-433C (DS) mutations (*Kwon et al., 2015*) and is a chimera of 426c gp120 and BG505 gp41 (*Joyce et al., 2017*). This glycan-depleted protein construct is henceforth referred to as 426c DS-SOSIP D3 (*Figure 1—figure supplement 1*). The VRC01$_{GL}$ construct comprises the germline VH gene reverted sequences of VH1-2*02, which includes CDRH1 and CDRH2 along with the mature CDRH3 of VRC01 and the germline VK3-11 with the mature CDRL3 of VRC01 (*Figure 1—figure supplement 2*). Initial complex formation was evaluated using negative-staining EM, which revealed sub-stoichiometric binding of VRC01$_{GL}$ Fab to 426c DS-SOSIP D3 (*Figure 1—figure supplement 3A*). The VRC01$_{GL}$ Fab appeared to have a much lower affinity for 426c DS-SOSIP D3, compared to the VRC01$_{GL}$ IgG (the latter bound with an apparent equilibrium dissociation constant of 43 nM (*Figure 1B*), neglecting the effect of avidity). We next attempted to enhance binding of VRC01$_{GL}$ Fab to 426c DS-SOSIP D3 by utilizing a mild glutaraldehyde cross-linking strategy. As expected, we observed significantly increased saturation of 426c DS-SOSIP D3 trimers by VRC01$_{GL}$ Fab, indicating covalent tethering following initial engagement of VRC01$_{GL}$ to the CD4$_{BS}$ was a suitable approach to enrich for and study VRC01$_{GL}$-bound complexes (*Figure 1—figure supplement 3A*). We therefore engineered a disulfide bond between G459C$_{gp120}$ (426c DS-SOSIP D3†) and the VRC01$_{GL}$ heavy chain A60C (denoted as 426c DS-SOSIP D3†-VRC01$_{GL}$) (*Figure 1—figure supplement 1* and *Figure 1—figure supplement 2*). This strategy was previously employed to enhance binding of VRC01$_{MAT}$ to SOSIP trimers without altering interface contacts (*Stewart-Jones et al., 2016*). Using this method, we purified an enriched fraction of Fab-bound trimers and used this sample for structural characterization (*Figure 1C*, *Figure 1—figure supplement 3A–D*). This strategy led us to determine two cryoEM reconstructions of the 426c DS-SOSIP D3†-VRC01$_{GL}$ complex (*Figure 1D–E*, *Figure 1—figure supplement 3E–H*, *Figure 1—figure supplement 4A–D*, *Figure 1—figure supplement 5*): one with three bound Fabs at 3.8 Å resolution (*Figure 1D*, *Table 1*, *Figure 1—figure supplement 3E*, *Figure 1—figure supplement 4A–B*), and one with two bound Fabs at 4.8 Å resolution (*Figure 1E*, *Figure 1—figure supplement 3H*, *Figure 1—figure supplement 4C–D*).

Similarly to what was reported for the B41 SOSIP trimer (*Ozorowski et al., 2017*), the 426c DS-SOSIP D3†-VRC01$_{GL}$ V1/V2 apex is closed under cryoEM conditions (*Figure 1D,E,F*) whereas it appears open in the conditions we used for negative-staining sample preparation (*Ozorowski et al., 2017*) (*Figure 1—figure supplement 3B*). We note that the closed 426c DS-SOSIP D3†-VRC01$_{GL}$ trimeric Env conformation observed in cryoEM lacks a formed gp120 bridging sheet, which is reminiscent of other closed SOSIP trimer structures (*Figure 1G*) (*Julien et al., 2015*; *Lyumkis et al., 2013b*; *Pancera et al., 2014*; *Stewart-Jones et al., 2016*). VRC01$_{GL}$-class antibodies have recently been shown to also bind to core gp120 constructs in the presence of a bridging sheet (*Scharf et al., 2016*). Our data reveal that VRC01$_{GL}$ could also bind a prefusion closed conformation, which had previously only been reported for its mature counterpart, VRC01$_{MAT}$ (*Stewart-Jones et al., 2016*) (*Figure 1G*).

### Structural analysis of the region surrounding the CD4$_{BS}$ in 426c DS-SOSIP D3

Removal of CD4$_{BS}$-surrounding carbohydrates has been shown to enhance binding of CD4$_{BS}$-targeted germline VRC01-class antibodies and to increase the antigenicity of this region (*McGuire et al., 2016*; *McGuire et al., 2013*; *Stamatatos et al., 2017*; *Zhou et al., 2017*). Our structural analysis reveals that the 426c DS-SOSIP naturally lacks an NLGS at position Asn234 near the CD4$_{BS}$, which is otherwise conserved in 80% of known circulating HIV-1

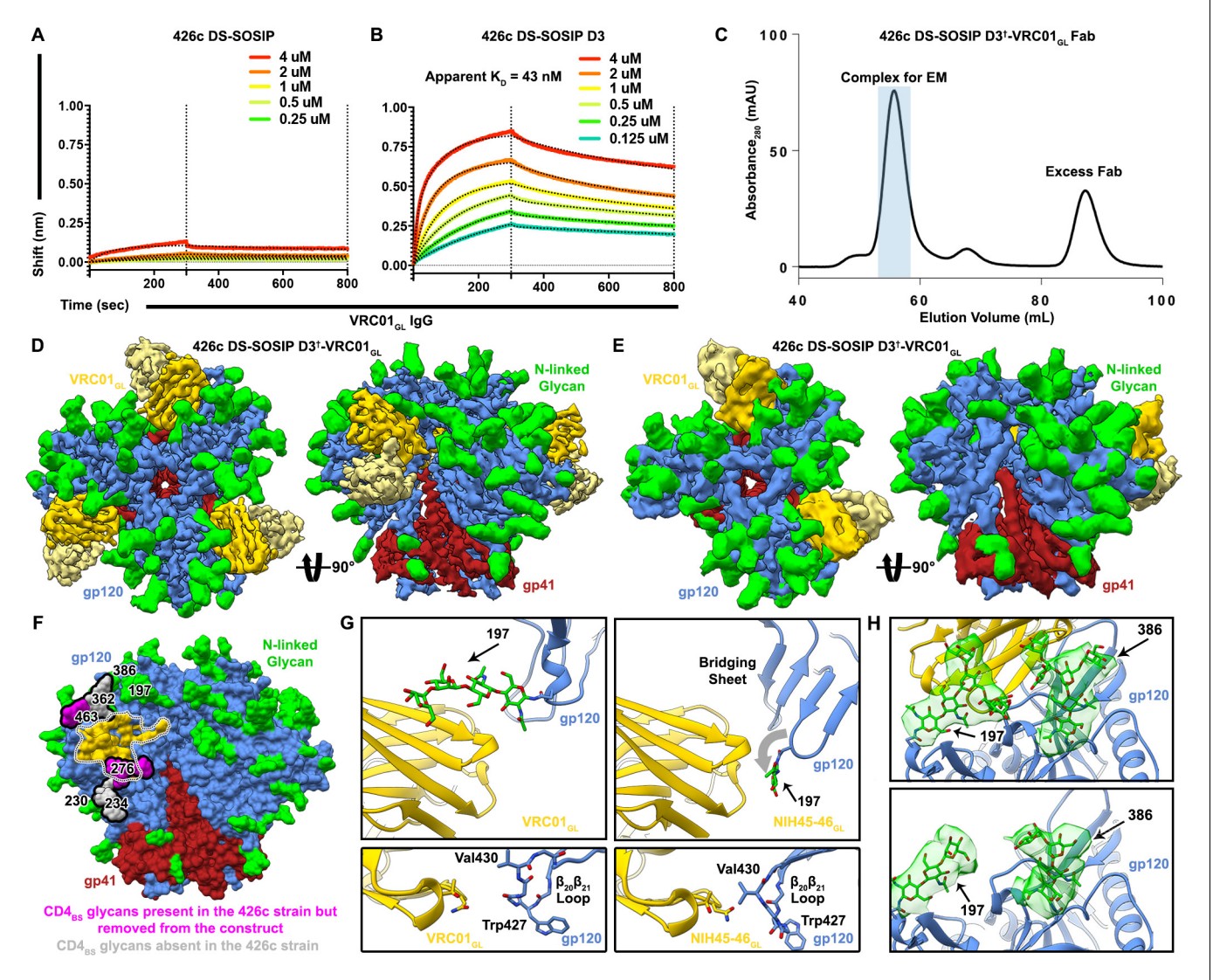

**Figure 1.** Structural characterization of the 426c DS-SOSIP D3†-VRC01GL complex. (**A–B**) BLI binding data of immobilized VRC01GL IgGs binding to WT 426c DS-SOSIP (**A**) or 426c DS-SOSIP D3 trimers. The concentrations of 426c DS-SOSIP trimers injected are indicated on each panel. Fit curves are colored as black dotted lines. A KD could not be determined in (**A**) due to the weak responses observed. The vertical dotted lines indicate the transition between association and dissociation phases. (**C**) Size-exclusion chromatogram of the purified 426c DS-SOSIP D3†-VRC01GL complex used for cryoEM structure determination. The pooled fractions used for cryoEM are highlighted in light blue. (**D**) Two orthogonal views of the 3.8 Å cryoEM reconstruction sharpened with a B-factor of −250 Å² whereas the glycan density is shown unsharpened. (**E**) Two orthogonal views of the asymmetric 4.8 Å reconstruction with two bound Fabs. (**F**) Surface representation of the 426c SOSIP trimer highlighting differences in glycosylation compared to the BG505 SOSIP. Glycans not present in 426c are colored light-gray and outlined. Glycans present in the 426c strain but removed by mutation from the 426c DS-SOSIP D3† construct are colored magenta and outlined. The gp120 surface buried at the interface with VRC01GL is indicated as a dotted outline and is colored yellow. (**G**) Comparison of the gp120 bridging sheet conformation when VRC01GL-class Fabs are bound to either 426c DS-SOSIP D3† trimer (*Top-left*) or a previously solved 426c gp120 core lacking selected NLGSs, such as the Asn276 NLGS (PDB: 5IGX) (*Top-right*). Comparisons of β20β21 loop conformations of each complex are shown below corresponding top panels. (**H**) Comparison of glycan density and position between VRC01GL-bound and VRC01GL-free protomers in the asymmetric cryoEM reconstruction shown in (**E**). (*Top*) Asn197 and Asn386 glycan density is stronger for protomers bound to VRC01GL Fab than for the gp120 protomer not bound to VRC01GL (*Bottom*). In panels D-H, gp120 protomers are shown in blue, gp41 in red, N-linked glycans in green and VRC01GL in dark and light yellow for the heavy and light chains, respectively.

DOI: https://doi.org/10.7554/eLife.37688.002

The following figure supplements are available for figure 1:

**Figure supplement 1.** Multiple sequence alignment of analyzed HIV-1 426c constructs.
DOI: https://doi.org/10.7554/eLife.37688.003

**Figure supplement 2.** Multiple sequence alignment of analyzed antibody and Fab constructs.

*Figure 1 continued on next page*

*Figure 1 continued*

DOI: https://doi.org/10.7554/eLife.37688.004

**Figure supplement 3.** Structural characterization of the 426c DS-SOSIP D3$^\dagger$-VRC01$_{GL}$ complex.

DOI: https://doi.org/10.7554/eLife.37688.005

**Figure supplement 4.** Validation of the 426c DS-SOSIP D3$^\dagger$-VRC01$_{GL}$ cryoEM reconstructions.

DOI: https://doi.org/10.7554/eLife.37688.006

**Figure supplement 5.** Comparison of gp120 interface contacts between VRC01$_{GL}$ and VRC01$_{MAT}$.

DOI: https://doi.org/10.7554/eLife.37688.007

**Figure supplement 6.** Example of glycans resolved in the 426c DS-SOSIP D3$^\dagger$-VRC01$_{GL}$ structure.

DOI: https://doi.org/10.7554/eLife.37688.008

strains (*Crooks et al., 2015*). Instead, 426c features a glycan at Asn230 that is more remote from the VRC01 epitope than glycan Asn234 (*Jardine et al., 2016b*) (*Figure 1F*). The oligosaccharide at position Asn230 appears to be highly dynamic, since only the two proximal N-acetyl-glucosamine (GlcNAc) moieties are resolved in the reconstruction (*Figure 1—figure supplement 6*) and does not interact with VRC01$_{GL}$ or other glycans in the complex. Previous structural characterization of clades A and G SOSIP trimers established that glycans at positions Asn276 and Asn234 are in close proximity to each other and likely restrain each others' conformational freedom (*Jardine et al., 2016b*; *Stewart-Jones et al., 2016*; *Zhou et al., 2017*). The absence of glycan Asn234 in 426c gp120 reduces local carbohydrate crowding near the CD4$_{BS}$ which could increase accessibility of this neutralization supersite (*Stewart-Jones et al., 2016*) and lead to altered local glycan processing (*Behrens et al., 2018*; *Bonomelli et al., 2011*).

The 426c DS-SOSIP also lacks an NLGS at position Asn362(363), which is present in 42% of strains deposited in the HIV database (*Gaschen et al., 2001*) (*Figure 1F*). This oligosaccharide is located distally from the viral membrane side of the SOSIP trimer (*Figure 1F*) and is sandwiched between the VRC01$_{MAT}$ heavy chain and glycan Asn386 in the structure of VRC01$_{MAT}$ bound to the JR-FL SOSIP trimer (clade B) (*Stewart-Jones et al., 2016*). Analysis of the asymmetric 426c DS-SOSIP D3$^\dagger$-VRC01$_{GL}$ structure, comprising two Fabs, revealed that Fab-bound protomers feature slightly better-resolved density for glycan Asn386 than the free protomer when visualized at the same contour level

**Table 1.** CryoEM data collection, refinement, and model validation statistics.

| Parameter | Value |
| --- | --- |
| Data Collection | |
| No. of Micrographs | 1993 |
| No. of Particles | 134,443 |
| Pixel size, Å | 1.36 |
| Defocus range, μM | 2.0–3.5 |
| Voltage, kV | 300 |
| Dose Rate, counts/pix/sec | 8 |
| Electron dose, e$^-$/Å$^2$ | 43 |
| Refinement | |
| Resolution, Å | 3.8 |
| Map-sharpening B factor, Å$^2$ | −230 |
| Model validation (3 Fab structure) | |
| Favored rotamers, % | 98.36% |
| Poor rotamers, % | 0.30% |
| Ramachandran outliers, % | 0.13% |
| Clash Score | 0.99 |
| Molprobity score | 1.02 |
| EM ringer score | 1.97 |

DOI: https://doi.org/10.7554/eLife.37688.009

(*Figure 1H*). These observations suggest that VRC01$_{GL}$ may stabilize the Asn386 glycan either through reduction of its conformational freedom and/or via direct interactions with the Fab framework region. The absence of glycan Asn362 or other topologically equivalent oligosaccharides in the 426c gp120 sequence likely contributes to increased accessibility of the CD4$_{BS}$ to VRC01$_{GL}$-class bnAbs due to the close proximity of this glycan to the epitope (*Stewart-Jones et al., 2016*) (*Figure 1F*).

Similarly to what is observed in available VRC01$_{MAT}$/SOSIP complex structures (*Stewart-Jones et al., 2016*), glycan Asn197 density is also strongest when bound to VRC01$_{GL}$, but appears weaker in the unbound protomer (*Figure 1H*), again indicating either Fab-induced stabilization or restriction of movement. The position of glycan Asn197 differs substantially between available structures of monomeric gp120 constructs bound to VRC01$_{GL}$-class Fabs and the VRC01$_{GL}$-bound SOSIP trimer reported here (*Figure 1G*) (*Scharf et al., 2016*). This variation in Asn197 positioning is guided by the formation of the gp120 bridging sheet in monomeric gp120, which would otherwise only form following CD4 receptor binding in the context of trimeric Env (*Figure 1G*) (*Kwon et al., 2012*; *Zhou et al., 2010*). This conformational difference includes the $\beta_{20}/\beta_{21}$ loop, whose orientation in the 426c DS-SOSIP D3$^{†}$-VRC01$_{GL}$ complex differs relative to crystal structures of VRC01$_{GL}$-class antibodies in complex with monomeric gp120 (*Figure 1G*) (*Scharf et al., 2016*). Although the $\beta_{20}/\beta_{21}$ loop is close to the VRC01 paratope, VRC01$_{MAT}$ was reported to have minimal preference in the conformation of the bridging sheet or $\beta_{20}/\beta_{21}$ region, as 87% of its contact surface area includes the conformationally invariant outer domain of gp120 (*Zhou et al., 2010*). Whether or not the conformation of the $\beta_{20}/\beta_{21}$ region directly impacts germline VRC01-class antibody binding affinities remains unclear. However, VRC01$_{GL}$ in complex with gp120 constructs lacking this domain have been determined (eOD-GT6 and eOD-GT8), demonstrating that these germline mAbs do not strictly require this region for CD4$_{BS}$ recognition when glycans surrounding the CD4$_{BS}$ are also removed (*Jardine et al., 2013*).

## Wild-type V5 loop NLGSs of the 426c core did not hinder binding to VRC01$_{GL}$ Fabs

One of the mechanisms by which HIV-1 Env has evolved to avoid detection by the progenitors of VRC01-class bnAbs is by selection of V5 loop NLGSs (*Huang et al., 2016*; *Li et al., 2011*; *Zhou et al., 2010*) which sterically limit access to the CD4$_{BS}$. The observation that VRC01$_{GL}$ may accommodate carbohydrates surrounding the CD4$_{BS}$ in our 426c DS-SOSIP D3$^{†}$-VRC01$_{GL}$ cryoEM structure prompted us to assess the effect on binding of the two V5 loop putative NLGSs mutated in 426c DS-SOSIP D3$^{†}$-VRC01$_{GL}$. With 426c Env trimers, we previously found that VRC01$_{GL}$ binding could be detected following removal of glycan Asn276, and was further enhanced following removal of wild-type NLGSs at positions Asn460 and Asn463 (*McGuire et al., 2013*). We also demonstrated that removing the V1/V2, and V3 loops in gp140 Env trimers further increased binding of multiple VRC01-class germline antibodies relative to trimers only containing glycan-depleting mutations (*McGuire et al., 2014*; *McGuire et al., 2016*). However, the effects of V1/V2 and V3 loop deletion on VRC01$_{GL}$ binding to 426c core constructs in the presence of glycans remains unclear. Here we reintroduced the two NLGSs at positions Asn460 and Asn463 (in the V5 loop) and assessed their individual and cumulative effects on VRC01$_{GL}$ engagement of the 426c core construct comprised of gp120 residues 44 to 492, and lacking the V1/V2 and V3 variable loops (*Kwon et al., 2012*). Expanding on our prior work (*McGuire et al., 2016*), the following four 426c core glycan-deleted combinations were tested using HEK293F-expressed protein constructs: S278A/T462A/T465A, S278A/T465A, S278A/T462A, and S278A (*Figure 1—figure supplement 1*, *Figure 2A–D*, *Supplementary file 1*).

Reintroduction of the NLGS at position Asn460 (S278A/T465A) in the 426c core had no detectable impact on VRC01$_{GL}$ binding affinity despite the predicted overlap of a putative carbohydrate at position Asn460 with the bound VRC01$_{GL}$ Fab (*Guo et al., 2012*) (*Figure 2A–B*, *Supplementary file 1*). Indeed, our previous work removing NLGSs at positions Asn460 and Asn463 via the N460D/N463D mutations had only a relatively minor impact on VRC01$_{GL}$ engagement to trimeric 426c constructs compared to the large increase in binding observed following removal of the native Asn276 NLGS (*McGuire et al., 2016*; *McGuire et al., 2013*). To better understand the molecular rationale of these observations with the monomeric 426c core construct, we engineered a disulfide-linked 426c core$^{†}$-VRC01$_{GL}$ complex containing all wild-type NLGSs (426c core$^{†}$-VRC01$_{GL}$). We co-

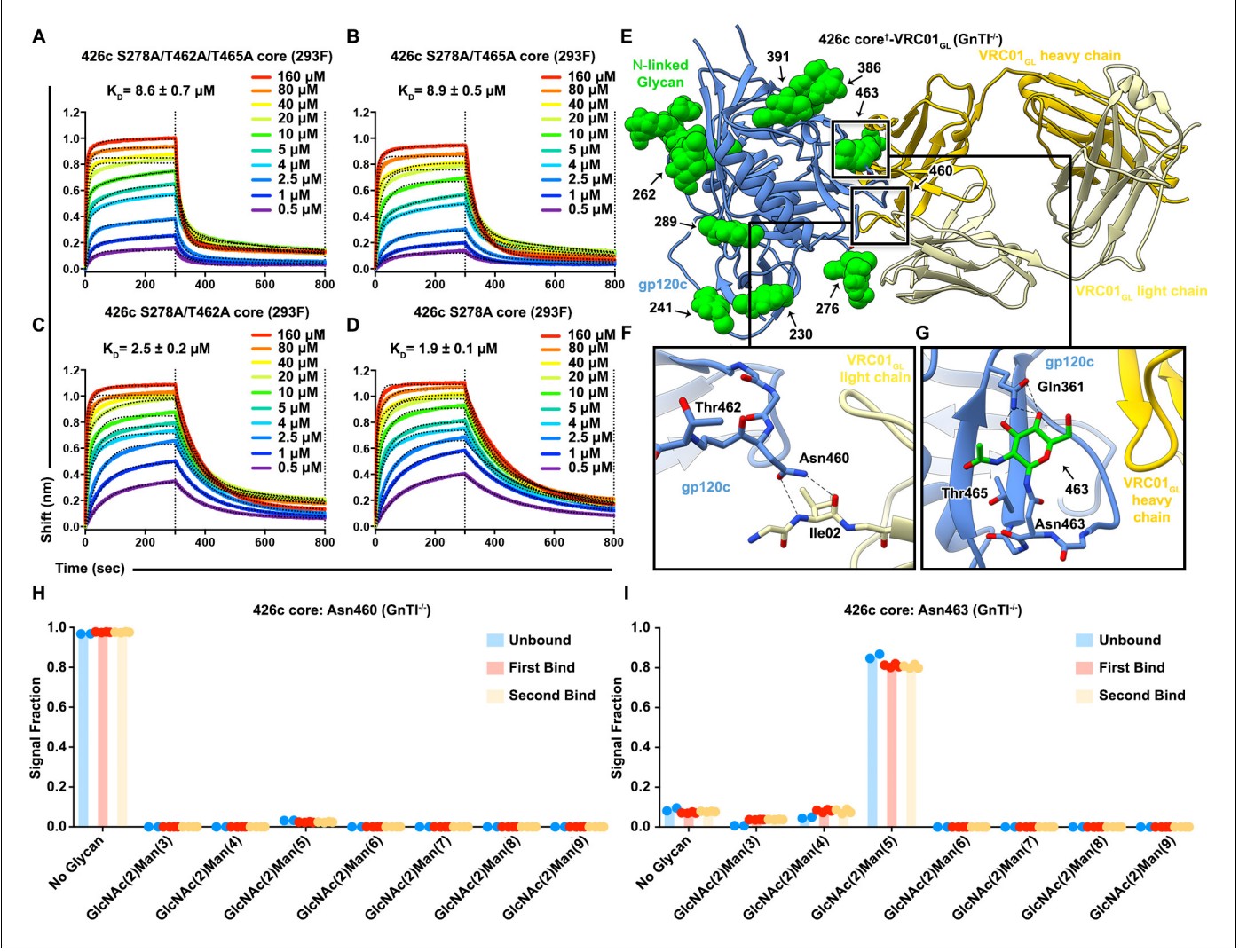

**Figure 2.** Reintroduction of V5 loop NLGSs does not hinder VRC01$_{GL}$ binding to the 426c core. (A–D) BLI curves and the corresponding equilibrium dissociation constants for VRC01$_{GL}$ IgG binding to the S278A/T462A/T465A (**A**), S278A/T462A (**B**), S278A/T465A (**C**), and S278A (**D**) 426c core constructs lacking either one or several glycans in the V5 and D loops. The concentrations of 426c core injected and the color key is indicated on each panel. Fitted curves are colored as black dotted lines. The vertical dashed lines indicate the transition between association and dissociation phases. (**E**) Ribbon diagram of the 426c core†-VRC01$_{GL}$ complex crystal structure. gp120 is colored blue, VRC01$_{GL}$ Fab is colored yellow (heavy chain: dark yellow; light chain: light yellow), and resolved gp120 glycans are shown in surface representation and colored green. (**F**) Close-up view of the gp120 Asn460 contacts with the backbone carbonyl and amide groups of the light chain VRC01$_{GL}$ residue Ile02. (**G**) Close-up view of the (GlcNAc)$_1$ at position Asn463 of gp120. Oligosaccharides are labeled by the corresponding Asn residue they are linked to. Hydrogen bonds are represented as dashed lines. (**H–I**) Semi-quantitative LC-MS/MS analysis of VRC01$_{GL}$-based IP experiments depicting the relative signal intensities for identified Asn460 (**H**) and Asn463 (**I**) glycoforms in unbound (blue), first binding event (red), and second binding event (yellow) fractions. The 'unbound' material indicates 426c core glycoforms that did not bind VRC01$_{GL}$ well following three binding steps. The 'first' binding event corresponds to 426c core elution fractions following collection of the sample flow-through and three rigorous wash cycles. The 'second' binding event follows a rebinding of the aforementioned flow-through, performing three additional washes, and eluting any residual bound material from the VRC01$_{GL}$ affinity column and collecting this fraction. Colored dots associated with their corresponding histogram bars represent individual values extracted from each experimental replicate, with the bar itself representing the experimental mean signal fraction.

DOI: https://doi.org/10.7554/eLife.37688.010

The following figure supplement is available for figure 2:

**Figure supplement 1.** Representative LC-MS/MS glycan identifications of 426c core constructs.
DOI: https://doi.org/10.7554/eLife.37688.011

expressed these proteins using HEK293 GnTI$^{-/-}$ cells, which lack N-acetyl-glucosaminyltransferase I activity and thus are unable to generate complex *N*-linked carbohydrates (*Wright and Morrison, 1994*). We then determined its crystal structure at 2.3 Å resolution after endoglycosidase H (EndoH) treatment to facilitate crystallization (*Depetris et al., 2012*; *Freeze and Kranz, 2010*) (*Figure 2E*, *Table 2*). Despite harboring an NLGS, no glycan density could be resolved at position Asn460 in either of the two molecules of the asymmetric unit. Instead, the Asn460 side chain is hydrogen bonded to the backbone amide and carbonyl groups of the VRC01$_{GL}$ light-chain residue, Ile02 (Ile02$_{LC}$) (*Kong et al., 2016*)(*Figure 2F*). In support of this observation, we detected only unglycosylated Asn460 peptide fragments when analyzing tryptic digests of this sample with liquid chromatography coupled to electron transfer/high-energy collision-dissociation tandem mass-spectrometry (LC-MS/MS) (*Figure 2H*, *Figure 2—figure supplement 1A,B*). Furthermore, only low levels of glycosylation were detected at Asn460 by qualitative LC-MS/MS analysis of unliganded 426c core (lacking the G459C mutation) (*Figure 2—figure supplement 1C*). Additionally, we performed VRC01$_{GL}$-based immunoprecipitation (IP) experiments utilizing a VRC01$_{GL}$ affinity column and the same 426c core construct. Semi-quantitative LC-MS/MS comparison of the 426c gp120 core samples from fractions that did not bind to VRC01$_{GL}$ ('unbound' flow-through), and those that did ('bound' elution),

**Table 2.** Crystallographic data collection and refinement statistics

| | 426c core$^{†}$-VRC01$_{GL}$ |
|---|---|
| **Data collection** | |
| Space group | C2 |
| Cell dimensions | |
| $a$, $b$, $c$ (Å) | 197.082, 109.003, 103.225 |
| $α$, $β$, $γ$ (°) | 90.000, 114.468, 90.000 |
| Resolution (Å) | 50–2.32 (2.36–2.32)* |
| $R_{sym}$ or $R_{merge}$ | 0.076 (0.643)* |
| $I/sI$ | 23.4 (1.8)* |
| Completeness (%) | 95.6 (66.8)* |
| Redundancy | 7.4 (5.7)* |
| CC1/2 | (0.823)* |
| **Refinement** | |
| Resolution (Å) | 46.98–2.315 (2.398–2.315)* |
| No. reflections | 83086 |
| $R_{work}/R_{free}$ | 24.38/29.55 (42.67/49.28) |
| No. atoms | 12470 |
| Protein | 11746 |
| Water | 325 |
| Ligand | 399 |
| B-factors (Å$^2$) | 74.22 |
| Protein | 73.57 |
| Water | 69.62 |
| Ligand | 97.10 |
| R.m.s deviations | |
| Bond lengths (Å) | 0.003 |
| Bond angles (°) | 0.60 |
| Ramachadran Favored % | 93.39 |
| Ramachadran Outliers % | 0.13 |
| MolProbity all-atoms clashscore | 4.05 |

DOI: https://doi.org/10.7554/eLife.37688.012

revealed no difference in glycan occupancy of the Asn460 NLGS (*Figure 2H*). This indicated this sequon is rarely glycosylated in 426c core and explains its negligible impact on VRC01$_{GL}$ binding. The predicted overlap of glycan Asn460 with VRC01$_{GL}$ and the absence of a resolved proximal GlcNAc in the crystal structure of 426c core$^{†}$-VRC01$_{GL}$ also suggests a likely strict preference for the unglycosylated Asn460 glycoform of the 426c gp120 for binding (*Figure 2F*).

We furthermore observed that reintroduction of the Asn463 NLGS (S278A/T462A or S278A) also did not result in a reduction in VRC01$_{GL}$ binding relative to the 426c S278A/T462A/T465A core (*Figure 2A,C,D*, *Supplementary file 1*). This result was unexpected, as V5 glycosylation of Env trimers containing all variable loops have been reported to negatively affect VRC01$_{GL}$ recognition of the CD4$_{BS}$ (*Huang et al., 2016*; *Li et al., 2011*; *McGuire et al., 2013*; *Zhou et al., 2010*). We observed electron density for the proximal GlcNAc linked to Asn463 in one of the two molecules of the asymmetric unit of the 426c core$^{†}$-VRC01$_{GL}$ crystal structure and cross-validated the presence of this post-translational modification using LC-MS/MS (*Figure 2G*). VRC01$_{GL}$-based IP experiments followed by semi-quantitative LC-MS/MS validated these structural observations by detecting an Asn463 glycosylation profile which was indistinguishable between 'bound' elution and 'unbound' flow-through fractions (*Figure 2I*). This supports our BLI data suggesting the Asn463 glycan does not hinder VRC01$_{GL}$ binding in the context of the 426c core and that this site is glycosylated.

## VRC01$_{GL}$ Fab bound to the Asn276 glycan-containing 426c core construct

A hallmark of VRC01-class bnAb maturation is the shortening of the CDRL1 loop length and/or the addition of glycine residues, both of which have been proposed to enable accommodation of the Asn276 glycan near the CD4$_{BS}$ (*Jardine et al., 2016a*; *Scharf et al., 2016*; *Wu et al., 2015*; *Zhou et al., 2010*). Although VRC01$_{MAT}$ was shown to bind to the trimeric Env CD4$_{BS}$ in the presence of glycan Asn276, its removal significantly increased binding affinity and neutralization potency (*Jardine et al., 2013*; *McGuire et al., 2016*; *McGuire et al., 2013*; *Medina-Ramírez et al., 2017a*; *Stamatatos et al., 2017*). Removal of the Asn276 NLGS from certain trimeric SOSIP constructs by either N276D or (S/T)278(A/R) mutations significantly enhanced the antigenicity of the VRC01 epitope (*McGuire et al., 2016*; *McGuire et al., 2013*). However, when such glycan-depleted trimeric Env constructs were used as immunogens, the antibodies they elicited failed to overcome the glycan present at position Asn276 of wild-type viruses (*Briney et al., 2016*; *Dosenovic et al., 2015*). Removal of glycan Asn276 through N276A substitution abrogated VRC01$_{GL}$ interactions, indicating this amino acid residue was critical for binding (*McGuire et al., 2016*). These observations suggest that initial engagement of VRC01-class bnAb precursors in infected individuals occurs with an asparagine at position 276 and may also be possible, at low levels, in the presence of a glycan at this NLGS (*Scharf et al., 2016*).

Considering the reduced glycan shielding of the 426c strain, the deletion of variable loops 1, 2, and 3 in our 426c core constructs, and the minimal impact V5 loop NLGSs had on VRC01$_{GL}$ binding, we tested whether the wild type 426c core construct could interact with VRC01$_{GL}$ in the presence of the Asn276 NLGS (*Figure 1—figure supplement 1*). BLI analysis revealed VRC01$_{GL}$ bound similarly to both the HEK293F and HEK293 GnTi$^{-/-}$-expressed 426c cores with equilibrium dissociation constants of 11 μM and 15 μM, respectively (*Figure 3A,B*). The crystal structure of the disulfide engineered 426c core$^{†}$-VRC01$_{GL}$ complex expressed in GnTi$^{-/-}$ cells further reveals the presence of a resolved glycan at position Asn276 in one of the two molecules of the asymmetric unit, indicating VRC01$_{GL}$ bound to both Asn276 glycosylated and unglycosylated species (*Figure 3C–J*).

Analysis of this structure highlights distinct sets of interactions observed between the light chains of VRC01$_{GL}$ or VRC01$_{MAT}$ and glycan Asn276, which is rotated ~90° when comparing the two structures (*Figure 3C–D*). In line with our previous observation that Asn276 is important for VRC01$_{GL}$ recognition (*McGuire et al., 2016*), we observed that Asn276 is hydrogen bonded to the VRC01$_{GL}$ light chain residue Tyr91 in our 426c core$^{†}$-VRC01$_{GL}$ and 426c DS-SOSIP D3$^{†}$-VRC01$_{GL}$ structures, but not in the JR-FL SOSIP-VRC01$_{MAT}$ crosslinked complex structure (*Figure 3C–E*) (PDB: 5FYK) (*Stewart-Jones et al., 2016*). The CDRL1 of VRC01$_{GL}$ has been shown to adopt multiple conformations both when unliganded and when bound to eOD-GT6; the latter of which lacked a glycan at position Asn276 (*Jardine et al., 2013*). The CDRL1 loop of NIH45-46$_{GL}$ (a germline VRC01-class antibody) bound to the 426c core TM4 adopts a similar orientation as the one observed for the CDRL1 loop of VRC01$_{GL}$ bound to eOD-GT6 (*Scharf et al., 2016*) (*Jardine et al., 2013*)

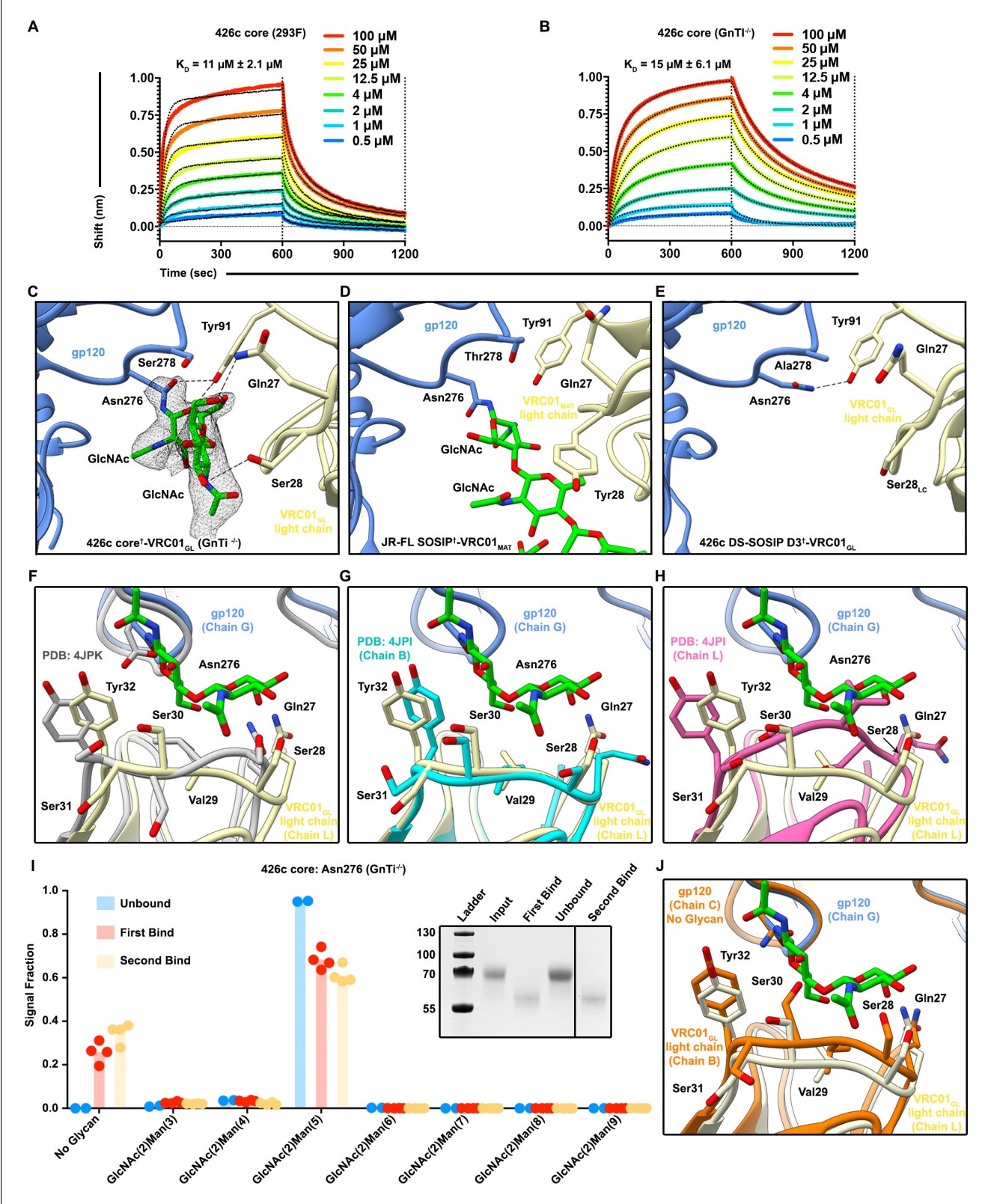

**Figure 3.** The VRC01$_{GL}$ Fab bound to the 426c core in presence of a glycan at position Asn276. (**A–B**) BLI binding data of the immobilized VRC01$_{GL}$ Fab with the 426c core expressed in either HEK293F (**A**) or HEK293 GnTI$^{-/-}$ cells (**B**). (**C**) Crystal structure of 426c core$^{†}$-VRC01$_{GL}$ highlighting glycan electron density at position Asn276 (grey mesh: 2F$_O$-Fc map contoured at 1.0σ) and amino-acid contacts for one molecule of the asymmetric unit. (**D**) Structure of VRC01$_{MAT}$ in complex (crosslinked) with the HIV-1 JR-FL SOSIP trimer (PDB ID: 5FYK) (*Stewart-Jones et al., 2016*) in the same orientation as in

*Figure 3 continued on next page*

Figure 3 continued

panel (C) and focusing on the glycan at position Asn276. (E) CryoEM structure of the 426c DS-SOSIP D3$^{\dagger}$-VRC01$_{GL}$ complex in the same orientation as in panel (C) and focusing on Asn276. Hydrogen bonds spanning 2.8–3.5 Å are depicted as dashed lines. (F–H) Comparison of VRC01$_{GL}$ CDRL1 conformations in the presence or absence of a glycan at position Asn276. In the three panels, gp120 is shown in blue cartoon representation and VRC01$_{GL}$ light chain in light yellow for our crystal structure of 426c core$^{\dagger}$-VRC01$_{GL}$. Residues Gln27 to Tyr32 of VRC01$_{GL}$ light chain are shown as sticks and labeled. (F) VRC01$_{GL}$ bound to eODGT6 (PDB ID: 4JPK)(*Jardine et al., 2013*) is shown in grey. (G) Chain B of unliganded VRC01$_{GL}$ (PDB ID: 4JPI) (*Jardine et al., 2013*) is shown in cyan. (H) Chain L of unliganded VRC01$_{GL}$ (PDB ID: 4JPI) (*Jardine et al., 2013*) is shown in pink. (I) Semi-quantitative LC-MS/MS analysis depicting the relative signal intensities for identified Asn276 glycoforms in unbound (blue), after the first binding event (red), and after the second binding event (yellow) fractions taken from VRC01$_{GL}$-based IP experiments. The 'unbound' material indicates 426c core glycoforms that did not bind VRC01$_{GL}$ following three binding events. Colored dots on corresponding histogram bars represent individual values extracted from each experimental replicate, with the bar itself representing the experimental mean signal fraction. (*Inset*) SDS-PAGE depicting the average molecular weight difference between wild-type 426c core species in 'unbound' flow-through and 'bound' elution fractions. (J) Structural comparison of VRC01$_{GL}$ CDRL1 conformations in the presence or absence of a glycan at position Asn276 in each of the molecules present in the asymmetric unit of our 426c core$^{\dagger}$-VRC01$_{GL}$ crystal structure.

DOI: https://doi.org/10.7554/eLife.37688.013

(*Supplementary file 2*). Comparisons between the structures of our 426c core$^{\dagger}$-VRC01$_{GL}$ complex, a putatively authentic germline VRC01/eOD-GT6 complex (PDB ID 4JPK [*Jardine et al., 2013*]) and the unliganded VRC01$_{GL}$ (PDB ID 4JPI) (*Jardine et al., 2013*), indicate that the CDRL1 of VRC01$_{GL}$ accommodates the Asn276 oligosaccharide in a conformation similar to the CDRL1 of unliganded VRC01$_{GL}$ (chain B) (*Supplementary file 2*)(*Jardine et al., 2013*), but differs from a complex with a gp120 core lacking this glycan (*Figure 3F–H*). This observation supports disulfide crosslinking of 426c core$^{\dagger}$-VRC01$_{GL}$ did not distort the binding interface into a non-native conformation for the accommodation of glycan Asn276 in our structure (*Figure 3C*). LC-MS/MS analysis of the sample used for crystallization revealed *N*-linked carbohydrates at position Asn276 of 426c core$^{\dagger}$-VRC01$_{GL}$ ranged from (GlcNAc)$_2$-(Man)$_4$ to (GlcNAc)$_2$-(Man)$_5$ (*Figure 4A,B*, *Figure 2—figure supplement 1E*). Unglycosylated Asn276 peptides were also identified, corroborating the presence of two populations of molecules in the crystal structure and the ability of VRC01$_{GL}$ to recognize both species (*Figure 4A*, *Figure 2—figure supplement 1F*).

Despite the EndoH treatment used to promote crystallization of 426c core$^{\dagger}$-VRC01$_{GL}$, no (GlcNAc)$_1$ glycopeptides were detected at the Asn276 NLGS by LC-MS/MS (*Figure 3A*). In contrast, we detected digested glycopeptides containing (GlcNAc)$_1$ moieties for other NLGSs, confirming the efficiency of the EndoH treatment (*Figure 2—figure supplement 1B*). These observations validated the Asn276 glycan density observed in the crystal structure and suggested that bound VRC01$_{GL}$ protected the glycan Asn276 from enzymatic digestion (*Figure 4A,B*), but not other oligosaccharides, such as glycan Asn463 (*Figure 2E,G* and *Figure 2—figure supplement 1B*). We further corroborated this hypothesis by analyzing EndoH-treated 426c core in the absence of co-expressed VRC01$_{GL}$ and confirmed the presence of (GlcNAc)$_1$ moieties at position Asn276 by LC-MS/MS (*Figure 3A,C*), supporting that digestion of this glycan was possible if not sterically hindered by the binding of this Fab (*Yet et al., 1988*).

To probe whether the disulfide cross-link promoted artificial accommodation of glycan Asn276, we performed an additional analysis with samples obtained from IP experiments using the 426c core construct lacking the G459C mutation. 426c core samples from the 'unbound' flow-through and 'bound' elution fractions had distinct migration profiles by SDS-PAGE (*Figure 3I*), with the bound fraction exhibiting higher electrophoretic mobility than the unbound species. LC-MS/MS revealed the bound fraction was enriched for unglycosylated Asn276 peptides, suggesting this subspecies was the preferred VRC01$_{GL}$ binder (*Figure 3I*). This result corroborates reports of VRC01$_{GL}$ binding occurring preferentially in the absence of a glycan at position 276 (*Jardine et al., 2013*; *McGuire et al., 2016*; *McGuire et al., 2013*; *Medina-Ramírez et al., 2017a*; *Scharf et al., 2016*; *Stamatatos et al., 2017*). However, we also detected that the majority of the Asn276 peptide signal was of the (GlcNAc)$_2$-(Man)$_5$ glycoform in bound fractions (approximately twice as much as the unglycosylated Asn276 signal) (*Figure 3I*), suggesting VRC01$_{GL}$ could indeed bind in the presence of this glycan and in the absence of an engineered cross-link. This experiment, along with our cross-linked 426c core$^{\dagger}$-VRC01$_{GL}$ crystal structure and qualitative LC-MS/MS, confirm both the glycosylated (*Figure 3C,F,G,H,I*) and unglycosylated Asn276 glycoforms are present following expression

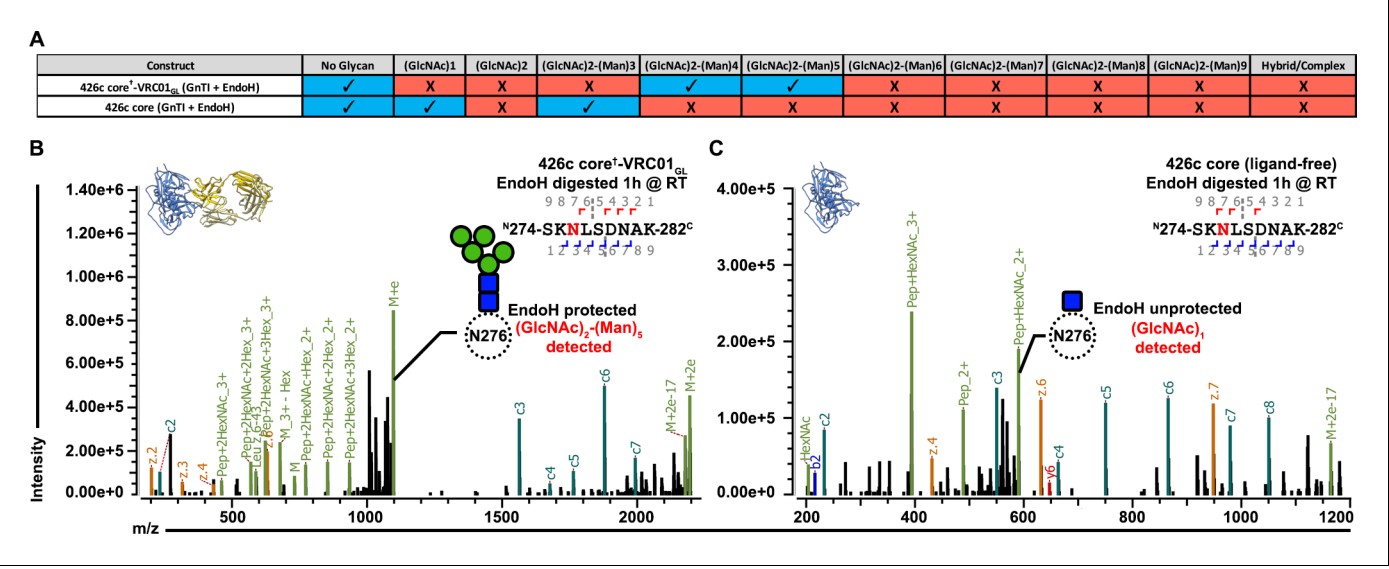

**Figure 4.** VRC01$_{GL}$ binding in the presence of a glycan at position Asn276 and protection against EndoH-mediated digestion. (**A**) Summary of identifications for 426c Asn276 glycopeptides. 426c core constructs that were subjected to qualitative LC-MS/MS are indicated on the left. Glycopeptide identifications detected using the Byonic software (*Bern et al., 2012*) are listed in blue and denoted with a check-mark (✓). (**B–C**) Representative LC-MS/MS spectra from panel (**A**) of glycan Asn276 identifications from the cross-linked 426c core†-VRC01$_{GL}$ and unliganded 426c core complex following EndoH digestion. The top-left ribbon diagram corresponds to the sample analyzed. The LC-MS/MS fragmentation pattern is indicated in the top-right inset. A graphical depiction of the Asn276 residue (dotted circle) and its associated identified glycan (blue: N-Acetylglucosamine, green : Mannose) are represented on the spectrum. The black line indicates identification of the precursor mass with neutral losses corresponding to the identified glycopeptide. Green peak labels correspond to precursor peptides with/without LC-MS/MS fragmentation occurring within the glycan. Red/orange peak labels represent identified x, y, and z fragments. Blue/teal peak labels highlight identified a, b, and c fragments. After EndoH digestion, a (GlcNAc)$_2$-(Man)$_5$ glycan was the predominant glycoform identified at position Asn276 with the sample used for crystallization (**B**) whereas a (GlcNAc)$_1$ glycan prevailed with the unliganded 426c core (**C**).

DOI: https://doi.org/10.7554/eLife.37688.014

and that VRC01$_{GL}$ could accommodate both. In summary, VRC01$_{GL}$ bound a 426c core with wild-type NLGSs, was sterically compatible with glycans present at positions Asn276 and Asn463, and strictly interacted with a subspecies of gp120 lacking a glycan at position Asn460.

## Modulation of glycan composition altered VRC01$_{GL}$ antibody recognition of the 426c core

Irrespective of the chosen expression system (HEK293F or HEK293 GnTI$^{-/-}$), our LC-MS/MS analyses showed that 426c core constructs all contained detectable levels of both the unglycosylated and the (GlcNAc)$_2$-(Man)$_5$ oligosaccharide variants at position Asn276 (*Figure 4A–B*, *Figure 3I*, *Figure 2—figure supplement 1D–G*). Since (GlcNAc)$_2$-(Man)$_5$ is a short glycan produced in mammalian cells (*Hossler et al., 2009*), and was the major detectable glycosylated form in VRC01$_{GL}$-based IP 'bound' elution fractions, we interrogated whether differential expression conditions known to enrich for (GlcNAc)$_2$-(Man)$_9$ glycans could negatively impact the binding of VRC01$_{GL}$ IgGs to 426c core. We thus compared VRC01$_{GL}$ binding to HEK293 GnTI$^{-/-}$-produced 426c core constructs expressed in the absence or presence of 100 µM kifunensine to yield a range of (GlcNAc)$_2$-(Man)$_5$ to (GlcNAc)$_2$-(Man)$_9$ glycans or to enrich for (GlcNAc)$_2$-(Man)$_9$ glycans, respectively (*Depetris et al., 2012*).

The efficacy of this strategy was confirmed by two orthogonal methods: (1) SDS-PAGE, which demonstrated the two expression conditions yielded samples with distinct migration profiles, and (2) LC-MS/MS, which confirmed enrichment for (GlcNAc)$_2$-(Man)$_9$ glycans in the presence of kifunensine (*Figure 5A*, *Figure 5—figure supplement 1A–C*). Importantly, binding affinities for the 426c core were improved by ~10 fold in the context of immobilized full-length VRC01$_{GL}$ IgGs relative to immobilized VRC01 Fabs. The 426c core expressed using HEK293 GnTI$^{-/-}$ in the absence of kifunensine bound VRC01$_{GL}$ IgG with a $K_D$ of 2 µM (*Figure 5—figure supplement 1B*, *Supplementary file 3*), whereas binding was significantly reduced when the 426c core was expressed in the presence of

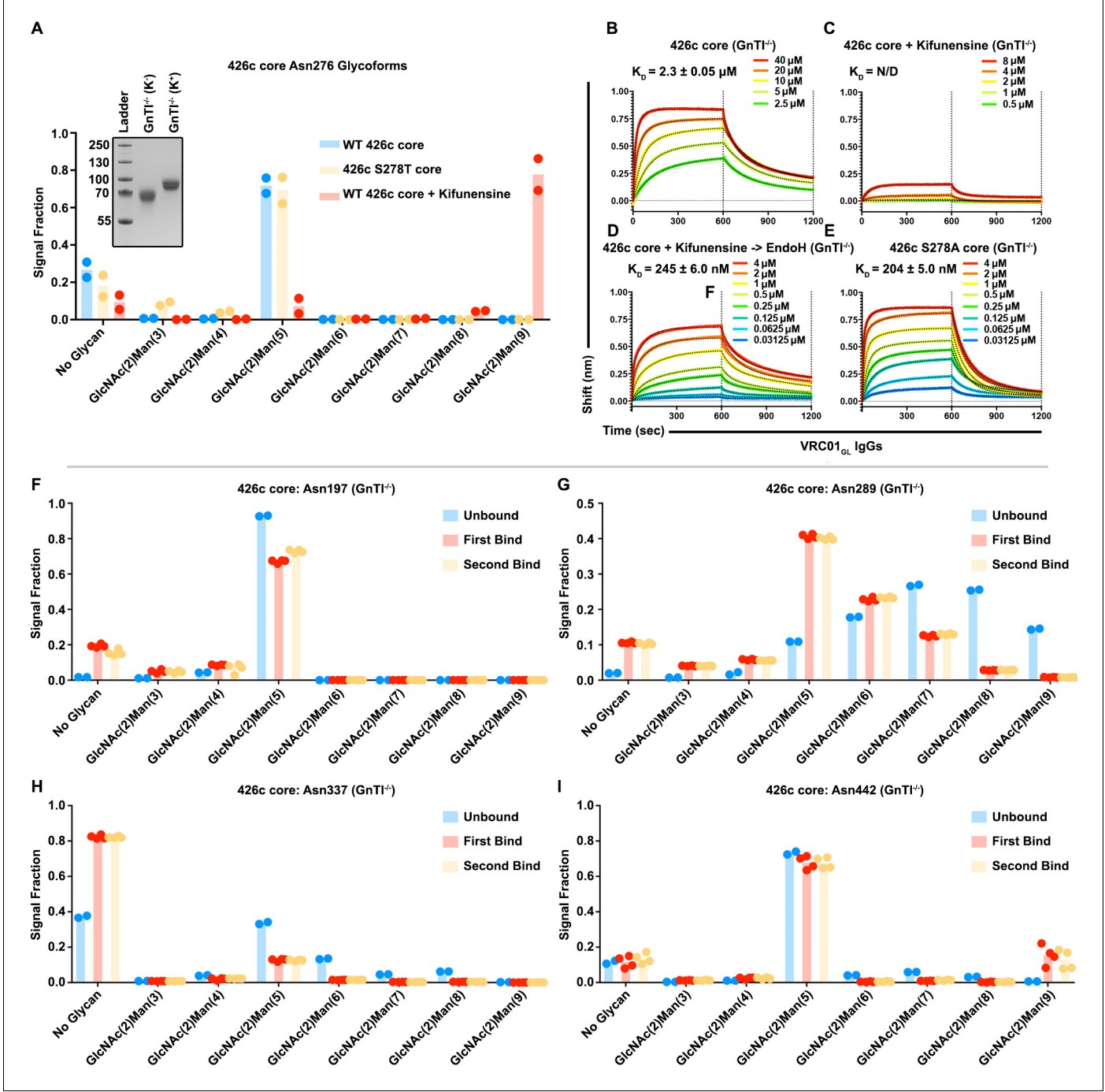

**Figure 5.** Glycan length impacted VRC01$_{GL}$ IgG recognition of the 426c core. (A) Semi-quantitative LC-MS/MS analysis depicting the relative signal intensities for identified Asn276 glycoforms in 426c core (blue), 426c S278T core (yellow), and 426c core expressed in the presence of 100 µM kifunensine (red). (*Inset*) SDS-PAGE demonstrating the molecular weight difference between the 426c core expressed in the absence (K$^-$) or presence (K$^+$) of 100 µM kifunensine. The molecular weights of the protein standards are indicated on the left. (B–E) BLI binding data and determined equilibrium dissociation constant values of VRC01$_{GL}$ IgG binding to the 426c core expressed using HEK293 GnTI$^{-/-}$ cells (B), the 426c core expressed using HEK293 GnTI$^{-/-}$ cells in the presence of 100 µM kifunensine (C), the 426c core expressed using HEK293 GnTI$^{-/-}$ cells and digested with EndoH (D), and the 426c S278A core expressed using HEK293 GnTI$^{-/-}$ cells (E). The concentrations of 426c core injected are indicated on each panel. Fitted curves are colored as black dotted lines. The vertical dotted lines indicate the transition between association and dissociation phases. N/D: not determined. (F–I) Semi-quantitative LC-MS/MS analysis depicting the relative signal intensities of unbound (blue), first binding event (red), and second binding event (yellow) fractions taken from VRC01$_{GL}$-based IP experiments. Glycoforms were analyzed for NLGSs Asn197 (G), Asn289 (H), Asn337 (I), and Asn442 (J). Colored

*Figure 5 continued on next page*

Figure 5 continued

dots in panels A and F-I represent individual values extracted from each experimental replicate, with the bar itself representing the experimental mean signal fraction.

DOI: https://doi.org/10.7554/eLife.37688.015

The following figure supplements are available for figure 5:

**Figure supplement 1.** LC-MS/MS glycan identifications of kifunensine-treated 426c core constructs.

DOI: https://doi.org/10.7554/eLife.37688.016

**Figure supplement 2.** Trimeric 426c DS-SOSIP has a variable glycan length at position Asn276.

DOI: https://doi.org/10.7554/eLife.37688.017

kifunensine (*Figure 5C*, *Supplementary file 3*). The VRC01$_{GL}$ IgG binding affinity to 426c core expressed in GnTI$^{-/-}$ cells in the presence of kifunensine was enhanced following treatment with EndoH ($K_D$ = 245 nM, *Figure 5D*, *Supplementary file 3*), confirming recognition of gp120 by VRC01$_{GL}$ IgGs can occur in the presence of a proximal GlcNAc at position Asn276. This VRC01$_{GL}$ IgG binding affinity was similar to the HEK293 GnTI$^{-/-}$-expressed 426c S278A core construct ($K_D$ = 204 nM), although the kinetics of binding differed for the two samples (*Figure 5E*).

Previous studies demonstrated enhanced VRC01$_{MAT}$ binding following glycan Asn276 removal, although VRC01$_{MAT}$ could also accommodate an Asn276 glycan when present (*Jardine et al., 2013*; *McGuire et al., 2013*; *Medina-Ramírez et al., 2017a*); *McGuire et al., 2016*; *Scharf et al., 2016*; *Stamatatos et al., 2017*; *Zhou et al., 2017*). Similarly to VRC01$_{MAT}$, VRC01$_{GL}$ binding was previously detected in the absence of the Asn276 glycan (*McGuire et al., 2016*; *McGuire et al., 2013*). In this present study, VRC01$_{GL}$ binding was also observed under typical expression conditions with the native Asn276 NLGS retained, but was reduced following expression in the presence of kifunensine (*Supplementary file 3*, *Figure 5A,B,C*, *Figure 5—figure supplement 1A,B,E,F*). This may indicate that kifunensine treatment results in more efficient glycosylation of some NLGSs or that differences in glycan length influence recognition of VRC01$_{GL}$-class antibodies to the CD4$_{BS}$, or both (*Figure 5A*). In line with these hypotheses, the VRC01$_{GL}$-based IP and subsequent semi-quantitative LC-MS/MS of the 426c core expressed in the absence of kifunensine revealed a consistent preference for short and/or unglycosylated species bound to VRC01$_{GL}$ (*Figure 3I*, *Figure 5F,G,H,I*). These findings explain the observed increase in electrophoretic mobility of 426c core species in VRC01$_{GL}$-based IP fractions that bound this antibody compared to the unbound fraction (*Figure 3I*).

While many 426c core glycans are likely to be affected by either kifunensine or EndoH treatment, the Asn276 oligosaccharide is expected to have a pronounced negative effect on VRC01$_{GL}$ binding due to its direct overlap with the VRC01$_{GL}$ epitope (*Stewart-Jones et al., 2016*; *Zhou et al., 2017*). The 426c core$^{†}$-VRC01$_{GL}$ crystal structure does not resolve ordered mannose rings at position Asn276, despite their high detected abundance by LC-MS/MS, and thus only the two proximal GlcNAc moieties were modeled in our structure (*Figure 4C*). This indicates the Asn276 mannose moieties are likely not directly involved in binding to VRC01$_{GL}$ light chain, but rather act as a steric barrier VRC01$_{GL}$ must overcome to interact with the CD4$_{BS}$. Qualitative LC-MS/MS analyses of the 426c DS-SOSIP trimer, expressed in the absence of kifunensine, revealed longer glycans at the Asn276 NLGS, which correlated with a poorer binding affinity, relative to the 426c core also expressed in the absence of kifunensine (*Figure 1A*, *Figure 5—figure supplement 2*). Indeed, interactions with the Asn276-linked mannose moieties might be restricted to VRC01$_{MAT}$, as these rings are well-resolved in corresponding crystal structures. VRC01$_{GL}$-specificity and compatibility for proximal GlcNAcs is made evident by the increased affinity of VRC01$_{GL}$ for the 426c core following digestion with EndoH (*Figure 5D*). These results indicate binding of several germline antibodies to gp120 core constructs could potentially be modulated by tailoring protein expression conditions, oligosaccharide length, and/or by endoglycosidase treatment, as opposed to strict mutations aimed at abolishing NLGSs (*Kong et al., 2010*).

## The amino acid sequence of an intact 426c core Asn276 NLGS modulated VRC01$_{GL}$ antibody recognition

Considering our prior work demonstrating that amino acid composition of the Asn276 NLGS impacted VRC01$_{GL}$ recognition independently of glycan presence (*McGuire et al., 2016*), and our

observation that VRC01$_{GL}$ could bind to 426c core in the presence of a native NLGS at position Asn276 (with a preferential selection for the unglycosylated variant), we decided to test whether altering the identity of the Asn276 NLGS at the 278 position could impact VRC01$_{GL}$-class antibody binding. Whereas 82% of sequenced HIV-1 clades harbor a threonine residue at position 278 of gp120, 426c gp120 contains a serine at this position. Since recent reports suggested NXT NLGS sites are more efficiently glycosylated relative to NXS sites (*Huang et al., 2017*), we compared the binding kinetics of full-length VRC01$_{GL}$-class IgGs, VRC01$_{GL}$ and 12A21$_{GL}$, to HEK293 GnTI$^{-/-}$-expressed 426c core and an 426c S278T core mutant (*Figure 1—figure supplement 1* and *Figure 1—figure supplement 2*). BLI experiments demonstrated the S278T mutation reduced the equilibrium dissociation constant by 10-fold relative to the wild type NLGS, mainly by decreasing association kinetics (*Figure 6A–D*, *Figure 5—figure supplement 2A*). The S278T mutation did not abrogate VRC01$_{GL}$-class antibody binding, despite an expected improvement in Asn276 glycosylation efficiency (*Huang et al., 2017*). Moreover, semi-quantitative LC-MS/MS revealed (GlcNAc)$_2$-(Man)$_5$ oligosaccharides were predominantly detected at position Asn276 in the S278T mutant, though some unglycosylated peptides were still present at low levels (*Figure 5A*, *Figure 2—figure supplement 1H,I*).

To rule out any potential impact residue identity at position 278 might have on VRC01$_{GL}$-class antibody binding affinities (irrespective of glycosylation status), we mutated the Asn276 NLGS using substitutions S278A, S278V, and S278R (*Figure 1—figure supplement 1*). Although the magnitude of

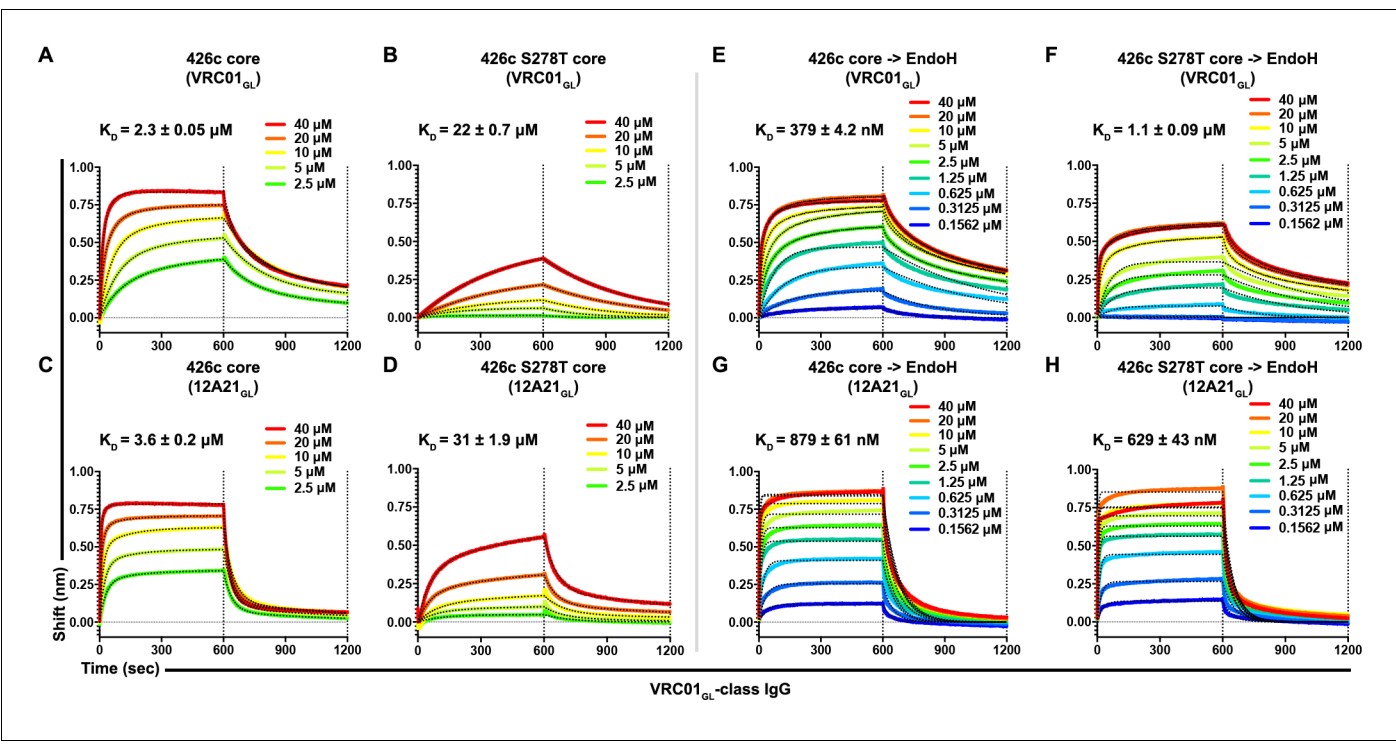

**Figure 6.** Asn276 glycosylation frequency modulated VRC01$_{GL}$-class IgGs recognition of the 426c core. (A–D) BLI binding data and associated K$_D$ values of 426c core constructs, expressed in HEK293 GnTI$^{-/-}$ cells, with two immobilized VRC01$_{GL}$-class IgGs. VRC01$_{GL}$ IgG binding was assessed against the 426c core (A) and the 426c S278T core (B). 12A21$_{GL}$ binding to the 426c core (C) and 426c S278T core (D) were also tested. (E–H) BLI binding data and corresponding K$_D$ values of 426c core constructs, expressed using HEK293 GnTI$^{-/-}$ cells and treated with EndoH, with VRC01$_{GL}$-class IgGs. VRC01$_{GL}$ IgG binding was assessed against the EndoH-treated 426c core (E) and the 426c S278T core (F). 12A21$_{GL}$ interactions with the EndoH-treated 426c core (G) and 426c S278T core (H) were also tested. The concentrations of 426c core injected and the color key are indicated on each panel. Fit curves are colored as black dotted lines. The vertical dotted lines indicate the transition between association and dissociation phases.
DOI: https://doi.org/10.7554/eLife.37688.018

The following figure supplement is available for figure 6:

**Figure supplement 1.** VRC01$_{GL}$ binding affinity was not significantly affected by the identity of the residue 278 in the absence of a glycan at position Asn276.
DOI: https://doi.org/10.7554/eLife.37688.019

binding was improved relative to both 426c core constructs containing a glycan at position Asn276, no appreciable difference in affinities was detected among any of these NLGS-depleted 426c core mutants (*Figure 6—figure supplement 1A–F*). Since binding was reduced following introduction of the S278T mutation, these results collectively suggest that a significant fraction of the interactions observed by BLI between the 426c core (containing an intact Asn276 NXS NLGS) and VRC01$_{GL}$ occurred with the unglycosylated Asn276 subspecies. This construct is expected to be less-frequently glycosylated at position Asn276 relative to the S278T mutant (*Huang et al., 2017*). However, VRC01$_{GL}$-class IgG binding was also detected with the 426c S278T core (*Figure 6A–D*, *Figure 6—figure supplement 1A*), and EndoH treatment (which retains core GlcNAcs) of both the S278 and S278T constructs significantly improved IgG binding relative to their untreated counterparts (*Figure 6E–H*, *Figure 6—figure supplement 1A*). Although these findings support that VRC01$_{GL}$-class binding is dampened by the Asn276 mannose moieties , interactions were still possible, as underscored by the presence of this carbohydrate in the crystal structure (*Figure 3C*), the EndoH protection assay (*Figure 4B,C*), and VRC01$_{GL}$-based IP LC-MS/MS analyses of non-crosslinked samples (*Figure 3I*). The ability of VRC01$_{GL}$-class antibodies to recognize a CD4$_{BS}$ containing an NLGS at position Asn276 and the linked oligosaccharide is unprecedented, and highlights a potentially unique feature of the 426c core construct containing all native NLGSs for engagement of VRC01$_{GL}$ antibodies.

## Discussion

Broad-spectrum and potent neutralization of HIV-1 by naturally occurring VRC01 bnAbs targeting the CD4$_{BS}$ is possible in humans (*Huang et al., 2016*). However, mature VRC01-class bnAbs are produced only in a small fraction of infected individuals and only after up to a decade following the initial infection (*Wu et al., 2015*). Due to the negligible binding of germline VRC01-class antibodies to 'wild-type' stabilized prefusion-closed SOSIP trimers, we sought out to understand putative mechanisms of primary engagement of this class of germline antibodies by various HIV-1 immunogens. To this end, we first engineered a disulfide bond between the glycan depleted 426c DS-SOSIP D3 and VRC01$_{GL}$ to promote complex formation. This tethering approach was recently described for VRC01$_{MAT}$ and led to native structures that do not suffer from any distortions (*Stewart-Jones et al., 2016*). It is unlikely that the engineered disulfide would force the VRC01$_{GL}$ and 426c gp120 to interact in a homogeneous way. Indeed, an engineered disulfide bond would only prevent dissociation of the two proteins (similarly to what we previously described for mature VRC01 [*Stewart-Jones et al., 2016*]), but will not force them to interact uniformly (*Stewart-Jones et al., 2016*). Using this crosslinking strategy, we demonstrated that VRC01$_{GL}$ bound to the deglycosylated 426c DS-SOSIP D3[†] trimer in the absence of a formed bridging sheet and that this binding stabilized surrounding CD4$_{BS}$ carbohydrates at position Asn197 and Asn386. Considering there was a minor enrichment for the unglycosylated Asn197 glycoform in 'bound' fractions following VRC01$_{GL}$-based immunoprecipitation of the 426c core, there could be an entropic cost associated with binding in the presence of this particular oligosaccharide. Additionally, we found that 426c naturally lacks glycans at positions Asn234 and Asn362, which likely enhanced accessibility of the CD4$_{BS}$ to bnAbs and could affect processing of nearby carbohydrates (*Behrens et al., 2018*), including glycan Asn276. We propose this is one of the reasons explaining the ability of 426c DS-SOSIP D3 and 426c core constructs to bind VRC01$_{GL}$-class antibodies.

Immunogen design efforts focusing on VRC01-class bnAbs have thus far largely relied on the mutagenic removal of NLGSs during priming followed by their reinsertion during subsequent boosts (*McGuire et al., 2013*; *Medina-Ramírez et al., 2017a*; *Zhou et al., 2017*). This strategy may not recapitulate the conditions of *bona fide* infections, as several CD4$_{BS}$ NLGSs are conserved amongst circulating HIV-1 strains (*Crooks et al., 2015*; *Lavine et al., 2012*; *Pritchard et al., 2015*; *Stewart-Jones et al., 2016*). The complete absence of these NLGSs during the priming phase may also remove a selection pressure guiding proper accommodation of these glycans during antibody affinity maturation, and could explain the limited success achieved thus far to elicit VRC01-class bnAbs capable of neutralizing natively glycosylated wild-type viruses. As a result, alternative priming and boost strategies retaining these native NLGSs may need to be considered.

Specifically, HIV-1 gp120 core constructs have previously been shown to have differential processing of NLGSs around the CD4$_{BS}$ relative to their trimeric SOSIP counterparts (*Bonomelli et al., 2011*). We observed that reintroduction of 426c gp120 V5 loop NLGSs did not negatively impact

VRC01$_{GL}$ binding to the 426c core in contrast with what was detected for trimeric 426c gp140 (*McGuire et al., 2013*). We found that VRC01$_{GL}$ Fabs and IgGs could bind to the 426c core (containing a wild-type Asn276 NLGS) but, as expected, not to a trimeric 426c DS-SOSIP construct containing all wild-type NLGSs (*Huang et al., 2016*). We put forward this interaction is likely permitted in the 426c core in part due to the absence of variable loops 1, 2, and 3, and could potentially be further augmented by the natural absence of glycans at position Asn234 and Asn362. It is also possible that stabilized trimeric pre-fusion closed SOSIP constructs impart additional negative steric effects not present in the monomeric gp120 core. This is supported by our observation that there is a reduced amount of trimming of the Asn276 glycan in the 426c DS-SOSIP trimer relative to the 426c core monomer. This could arise from steric properties of the trimer which would be absent in soluble monomeric constructs, thereby limiting unfettered access of the glycosylation machinery to this site during expression. It also remains to be determined if the conformation of the gp120 bridging sheet influences the recognition efficacy of this germline antibody.

We found that VRC01$_{GL}$ binding to 426c core constructs preferentially occurred using expression systems allowing for production of short glycans and provided a structural framework for accommodation of the Asn276 oligosaccharide by the VRC01$_{GL}$ CDRL1. As is the case for VRC01$_{MAT}$ (*Jardine et al., 2013*; *McGuire et al., 2016*; *McGuire et al., 2013*; *Medina-Ramírez et al., 2017a*; *Stamatatos et al., 2017*), we would like to emphasize that the unglycosylated Asn276 variant is the preferred VRC01$_{GL}$ binding partner and likely had an important contribution to the interactions observed by BLI. However, we detected glycosylation at position Asn276, both structurally and by LC-MS/MS following VRC01$_{GL}$-based IP experiments, in the VRC01$_{GL}$-bound fractions, in the presence and absence of cysteine cross-linking, respectively. This indicated accommodation of this oligosaccharide could occur during VRC01$_{GL}$ binding in the absence of CDRL1 loop shortening and/or glycine addition, arguing in favor of its retention in epitope-based constructs derived from the 426c strain of HIV-1.

The structural and biophysical data reported here provide a foundation for understanding how bnAbs targeting the HIV-1 CD4$_{BS}$ may be elicited in humans in the presence of a native Asn276 NLGS. We demonstrated structurally that VRC01$_{GL}$ binding to a 426c core is possible and appears to occur both in the absence and presence of a glycan at position Asn276, supporting recently proposed hypotheses (*Jardine et al., 2016b*; *Scharf et al., 2016*). Introduction of an NXS NLGS at position Asn276 of g120 core constructs lacking selected variable loops and in absence of glycans Asn234, Asn362(363), and Asn460 could prove a useful strategy to elicit VRC01-class antibodies during the priming phase of immunization. Additionally, the stark differences in binding observed between VRC01$_{GL}$ and 426c DS-SOSIP or 426c core constructs suggests 'germline-targeting' vaccine design strategies starting with a gp120 immunogen may be a promising alternative to current priming strategies focusing on glycan-depleted HIV-1 SOSIP trimers. Indeed, a remarkable success using site-directed epitope-based immunogens targeting other antigenic regions of HIV-1 Env has been recently achieved (*Xu et al., 2018*). In the context of VRC01$_{GL}$-targeted immunogens, the use of insect cell expression systems may further benefit recognition of the HIV-1 CD4$_{BS}$ due to the abundance of paucimannose sugars (*Altmann et al., 1999*). This approach has already been reported to enhance the immunogenicity of the CD4$_{BS}$ for other HIV-1 antibodies, but has yet to be specifically shown for antibodies of the VRC01 lineage (*Kong et al., 2010*). We highlight here the advantages of using an HIV-1 426c core construct for enhancing VRC01-class germline antibody binding relative to the glycan-depleted 426c trimeric Env SOSIP construct and propose that these observations be considered in future HIV-1 vaccine design efforts.

## Materials and methods

**Key resources table**

| Reagent type (species) or Resource | Designation | Source or reference | Identifiers | Additional information |
|---|---|---|---|---|
| Software, algorithm | Leginon | doi: 10.1016/ j.jsb.2005.03.010 | | http://emg.nysbc.org/redmine /projects/leginon/wiki/ Leginon_Homepage |

*Continued on next page*

*Continued*

| Reagent type (species) or Resource | Designation | Source or reference | Identifiers | Additional information |
|---|---|---|---|---|
| Software, algorithm | RELION-2 | doi: 10.1016/ j.jsb.2012.09.006 | RRID:SCR_016274 | http://www2.mrc-lmb. cam.ac.uk/relion/index .php/Main_Page |
| Software, algorithm | MotionCor2 | doi:10.1038/ nmeth.4193 | | http://emg.nysbc.org/ redmine/projects/appion /wiki/Appion_Home |
| Software, algorithm | GCTF | doi: 10.1016/ j.jsb.2015.11.003 | RRID:SCR_016500 | https://www.mrc-lmb. cam.ac.uk/kzhang/ |
| Software, algorithm | CTFFIND4 | doi:10.1016/ j.jsb.2015.08.008 | | http://grigorieflab. janelia.org/ctffind4 |
| Software, algorithm | Frealign | doi: 10.1016/ bs.mie.2016.04.013 | | http://grigorieflab. janelia.org/frealign |
| Software, algorithm | Appion Package | doi: 10.1016/ j.jsb.2009.01.002 | | http://emg.nysbc.org/ redmine/projects/appion /wiki/Appion_Home |
| Software, algorithm | DoG Picker | doi:10.1016/ j.jsb.2009.01.004 | | http://emg.nysbc.org/ redmine/projects/appion /wiki/Appion_Home |
| Software, algorithm | Coot | doi: 10.1107/ S0907444910007493 | RRID:SCR_014222 | http://www2.mrc-lmb.cam .ac.uk/Personal/pemsley/coot/ devel/build-info.html |
| Software, algorithm | Rosetta | doi: 10.1146/annurev. biochem.77.062906 .171838 | RRID:SCR_015701 | https://www.rosetta commons.org/software |
| Software, algorithm | UCSF Chimera | doi: 10.1002/jcc.20084 | RRID:SCR_004097 | http://plato.cgl.ucsf.edu /chimera/ |
| Software, algorithm | PMI-Byonic | doi: 10.1002/ 0471250953.bi1320s40 | | https://www.proteinmetrics .com/products/byonic/ |
| Software, algorithm | Skyline | doi: 10.1093/ bioinformatics/btq054 | RRID:SCR_014080 | https://skyline.ms/project /home/software/ Skyline/begin.view |
| Software, algorithm | Octet Data Acquisition | Pall ForteBio | CFR 10.0.3.12d | https://www.fortebio. com/octet-software.html |
| Software, algorithm | Octet Data Analysis | Pall ForteBio | CFR 10.0.3.1 | https://www.fortebio. com/octet-software.html |
| Software, algorithm | Phaser | doi:10.1107/ S0021889807021206 | | https://www.phenix-online. org/documentation/ reference/phaser.html |
| Software, algorithm | Phenix.refine | doi:10.1107/ S0907444912001308 | | https://www.phenix-online. org/documentation/ reference/refinement.html |
| Software, algorithm | GraphPad Prism | GraphPad | RRID:SCR_002798 | https://www.graphpad. com/scientific-software /prism/ |
| Software, algorithm | Pymol | Delano, 2002 | | https://pymol.org/2/ |
| Cell Line (*Homo sapiens*) | HEK293S GnTI-/- | ATCC | ATCC: CRL-3022; RRID:CVCL_A785 | https://www.atcc.org/ Products/All/CRL-3022.aspx |
| Cell Line (*Homo sapiens*) | HEK293F | ThermoFisher Scientifc | Cat# R79007 | https://www.thermofisher .com/order/catalog/ product/R79007 |
| Gene (*Homo sapiens*) | gl VRC01 Igk(3-11) | doi: 10.1126/ science.1192819 | | |
| Gene (*Homo sapiens*) | gl VRC01 Igg Fab | doi: 10.1038/ ncomms10618 | | |

*Continued on next page*

Continued

| Reagent type (species) or Resource | Designation | Source or reference | Identifiers | Additional information |
|---|---|---|---|---|
| Gene (*Homo sapiens*) | gl VRC01 Igg Fab A60C/C98S | This paper | | |
| Gene (*Homo sapiens*) | gl VRC01 Igg | 10.1126/science.1192819 | | |
| Gene (*Homo sapiens*) | gl 12A21 Igk(1-33) | 10.1084/jem.20122824 | | |
| Gene (*Homo sapiens*) | gl 12A21 Igg Fab | 10.1038/ncomms10618 | | |
| Gene (*Homo sapiens*) | gl 12A21 Igg | 10.1084/jem.20122824 | | |
| Strain, strain background (Human Immunodeficiency Virus-1, Strain: 426 c) | WT_426 c_DS-SOSIP | 10.1016/j.cell.2016.07.029 | | |
| Strain, strain background (Human Immunodeficiency Virus-1, Strain: 426 c) | 426 c_G459C_DS-SOSIP_D3 | This paper | | |
| Strain, strain background (Human Immunodeficiency Virus-1, Strain: 426 c) | 426 c_WT_gp120c_core | This paper | | |
| Strain, strain background (Human Immunodeficiency Virus-1, Strain: 426 c) | 426 c_G459C_gp120c_core | This paper | | |
| Strain, strain background (Human Immunodeficiency Virus-1, Strain: 426 c) | 426 c_S278A_gp120c_core | This paper | | |
| Strain, strain background (Human Immunodeficiency Virus-1, Strain: 426 c) | 426 c_S278A_T462A_gp120c_core | This paper | | |
| Strain, strain background (Human Immunodeficiency Virus-1, Strain: 426 c) | 426 c_S278A_T465A_gp120c_core | This paper | | |
| Strain, strain background (Human Immunodeficiency Virus-1, Strain: 426 c) | 426 c_S278A_T462A_T465A_gp120c_core | This paper | | |
| Strain, strain background (Human Immunodeficiency Virus-1, Strain: 426 c) | 426 c_S278T_gp120c | This paper | | |
| Recombinant DNA reagent | pTT3 | PMID: 11788735 | | https://biochimie.umontreal.ca/en/department/professors/yves-durocher/ |
| Recombinant DNA reagent | pVRC8400 | NIH | | |
| Chemical compound, drug | Kifunensine | Sigma-Aldrich | CAS Number 109944-15-2 | https://www.sigmaaldrich.com/catalog/product/sigma/k1140?lang=en®ion=US&gclid=Cj0KCQjwr53OBRCDARIsAL0vKrNtYwTyRzHU65HyVBwdntcP3kGpZ0ElVwYeSK3OcorLn0wf8U1iMQgaAssSEALw_wcB |

*Continued on next page*

*Continued*

| Reagent type (species) or Resource | Designation | Source or reference | Identifiers | Additional information |
|---|---|---|---|---|
| Peptide, recombinant protein | Endoglycosidase-H | New England Biolabs | Catalog #P0702S | https://www.neb.com/products/p0702-endo-h#Product%20Information |

## Protein purification

HEK293S GnTI$^{-/-}$ and HEK293F cell lines were used for protein expression. Both cells lines were authenticated using STR profiling. Mycoplasma tests were performed using the MycoProbe kit from R and D Systems and the samples were negative for contamination.

### 426c core$^{†}$-VRC01$_{GL}$

426c core$^{†}$-VRC01$_{GL}$ was expressed using HEK293S GnTI$^{-/-}$ cells . Cells were cultured in suspension and transfected with equal parts of 426c G459C core, VRC01$_{GL}$-A60C$_{HC}$, and VRC01$_{GL}$ light chain plasmids (500 µg total/L) using 293 Free Transfection Reagent (Novagen). After 6 days, cells were centrifuged at 4,500 rpm for 20 min and supernatant was filter-sterilized. A His-tag on the Fab heavy chain was utilized for purification by suspending His60 Ni-Superflow Resin (Takata) in the supernatant at 4°C overnight. The Ni resin was washed with a solution of 150 mM NaCl, 20 mM Tris pH 8.0, 20 mM Imidazole pH 7.0 and eluted with a solution of 300 mM NaCl, 50 mM Tris pH 8.0, 250 mM Imidazole pH 7.0. The sample was further purified by Size-exclusion chromatography (SEC) using a HiLoad 16/600 Superdex 200 pg (GE) column removing non-specific proteins and excess Fab. Fractions containing the complex were concentrated and treated with an excess of EndoH for one hour at 37°C. The complex was then rerun over an SEC S200 column and concentrated to ~10 mg/mL for crystallization trials.

### 426c DS-SOSIP D3$^{†}$-VRC01$_{GL}$

HEK293S GnTI$^{-/-}$ cells were transfected with 426c DS-SOSIP D3$^{†}$, VRC01$_{GL}$-A60C heavy chain, VRC01$_{GL}$ light chain and furin plasmids at a ratio of 3:1:1:1, as described above and previously (*Stewart-Jones et al., 2016*). Complexes were purified by the His-tag on the VRC01$_{GL}$ fragment as described above. Complexes were further purified on SEC as previously described and the peak containing both SOSIP and VRCO1$_{GL}$ were concentrated for cryoEM.

### 426c DS-SOSIP variants

426c DS-SOSIP D3 (non-crosslinked G459 variant with C-terminal strep-his tag) was expressed using HEK293F cells by co-transfection of 426c DS.SOSIP D3 and furin plasmids at a 5:1 ratio. The cells were not tested for mycoplasma contamination. 426c DS-SOSIP D3 was purified first by Ni-affinity and then by Streptactin-affinity, followed by enzymatic digestion and separation of the cleaved tag from the trimer by SEC using a HiLoad 16/600 Superdex 200 pg (GE).

### 426c core constructs

All 426c core constructs were expressed by the transfection protocol described above. Agarose Bound Galanthus Nivalis Lectin (Vector) was used to separate the cores from the supernatant. The resin was washed with 20 mM Tris pH 7.5, 100 mM NaCl, 1 mM EDTA and elution used a buffer containing 20 mM Tris pH 7.5, 100 mM NaCl, 1 mM EDTA, and 1M Methyl α-D-mannopyranoside. Samples were further purified by SEC using a HiLoad 16/600 Superdex 200 pg (GE) column.

### Antibodies

All antibodies were expressed by the transfection protocol described above using equal ratios of heavy and light chain encoding plasmids. Protein A Agarose (Pierce) resin was used to separate IgG from the supernatant. Protein A beads were washed with phosphate buffer saline (PBS) and elution used a commercially available IgG elution buffer at pH 2.0 (Pierce). Samples were buffer exchanged into PBS.

## Biolayer interferometry

BLI assays were performed with an Octet Red 96 instrument (ForteBio, Inc, Menlo Park, CA) at 29°C with shaking at 500 r.p.m. All measurements were corrected by subtracting the background signal obtained from duplicate traces generated with an irrelevant negative control IgG or Fab. For standard BLI assays, IgGs were immobilized on anti-AHC biosensors (at 20 μg/ml in PBS), or Fabs on anti-human Fab-CH1 (FAB2G, ForteBio) biosensors (at 40 μg/ in PBS), for 240 s. Sensors were then incubated for 1 min in kinetic buffer (KB: 1X PBS, 0.01% BSA, 0.02% Tween 20% and 0.005% $NaN_3$) to establish the baseline signal (nm shift). Antibody-loaded sensors were then immersed into solutions of purified recombinant samples for kinetic analysis. Analyses of DS-SOSIP trimers and 426c core constructs was performed by BLI using VRC01$_{GL}$ IgG and an extensive dilution series to determine accurate $K_D$ estimates. Samples expressed in the presence of 100 μM kifunensin and EndoH-digested were first buffer exchanged into PBS prior to dilution and kinetic analyses. Curve fitting to determine relative apparent antibody affinities for the samples was performed using a 1:1 binding model and the ForteBio data analysis software. Mean $k_{on}$, $k_{off}$, and $K_D$ values were determined by averaging all binding curves within a dilution series having $R^2$ values of greater than 95% confidence level.

## Crystallization

Crystallization conditions for the 426c core[†]-VRC01$_{GL}$ were screened using a Mosquito (ttplabtech)-dispensing robot. Screening was done with Rigaku Wizard Precipitant Synergy block no. 2, Molecular Dimensions Proplex screen HT-96, and Hampton Research Crystal Screen HT using the vapor diffusion method. Initial crystals were further optimized with Hampton Research Additive Screen to grow large and well-diffracting crystals. Final crystals were grown in a solution of 0.09M $MgCl_2$, 0.09M Na-Citrate pH 5.0, 13.5% PEG 4000, 0.1M $LiCl_2$. Crystals were cryoprotected in solutions containing 30% molar excess of their original reagents and 20% glycerol. Crystals diffracted to 2.3 Å. Data was collected at ALS 5.0.2 and processed using HKL2000 (*Otwinowski and Minor, 1997*).

## Structure solution and refinement

The structure of 426c core[†]-VRC01$_{GL}$ Fab was solved through molecular replacement using Phaser in CCP4 (*Collaborative Computational Project, Number 4, 1994*). The structure was further refined with COOT (*Emsley and Cowtan, 2004*) and Phenix (*Adams et al., 2010*). The refinement statistics are summarized in *Table 1*.

## Negative-stain EM sample preparation

All 426c DS-SOSIP constructs in this study (3 μL) were negatively stained at a final concentration of 0.008 mg/mL using Gilder Grids overlaid with a thin layer of carbon and 2% uranyl formate as previously described (*Veesler et al., 2014*).

## Negative-Stain EM data collection and processing

Data were collected on an FEI Technai 12 Spirit 120kV electron microscope equipped with a Gatan Ultrascan 4000 CCD camera. A total of 150–300 images were collected per sample by using a random defocus range of 1.1–2.0 μm with a total exposure of 45 e−/A$^2$. Data were automatically acquired using Leginon (*Suloway et al., 2005*), and data processing was carried out using Appion (*Lander et al., 2009*). The parameters of the contrast transfer function (CTF) were estimated using CTFFIND4 (*Mindell and Grigorieff, 2003*), and particles were picked in a reference-free manner using DoG picker (*Voss et al., 2009*). Particles were extracted with a binning factor of 2 after correcting for the effect of the CTF by flipping the phases of each micrograph with EMAN 1.9 (*Ludtke et al., 1999*). The 426c DS-SOSIP D3[†]-VRC01$_{GL}$ stack was pre-processed in RELION/2.1 (*Kimanius et al., 2016*; *Scheres, 2012b*; *Scheres, 2012a*) with an additional binning factor of 2 applied, resulting in a final pixel size of 6.4 Å. Resulting particles were sorted by reference-free 2D classification over 25 iterations. The best particles were chosen for 3D classification into six classes using RELION/2.1 (*Kimanius et al., 2016*). C3 symmetry was applied for 426c DS-SOSIP D3[†]-VRC01$_{GL}$, with the best 3D classes refined further in RELION/2.1 (*Kimanius et al., 2016*) using the gold-standard approach.

## CryoEM sample preparation

We applied 2 µL of 0.7 mg/mL of DS-SOSIP D3$^\dagger$-VRC01$_{GL}$ in 10 mM HEPES pH 7.5, 50 mM NaCl, 0.085 mM dodecyl-maltoside to glow-discharged C-flat CF-1.2/1.3–4 C-T-grids. Vitrification was performed by using an FEI Vitrobot Mark IV, using a blot time of 6 s at a temperature of 22°C and 100% humidity.

## CryoEM data collection

Data collection was performed automatically using Leginon (*Suloway et al., 2005*) to control an FEI Titan Krios Electron Microscope equipped with a Gatan Quantum GIF energy filter and a K2 Summit direct electron detector(*Li et al., 2013*) operating in electron-counting mode spanning a random defocus range between 2.0 and 3.5 µm. Approximately 2000 micrographs were collected with a pixel size of 1.36 Å at a dose rate of 8 counts per pixel per second and 15 s acquisition time (0.2 frame per second), yielding a final measured dose of 43 e$-$/Å$^2$ per movie.

## CryoEM data processing

Alignment of movie frames was carried out using MotionCor2 (*Zheng et al., 2017*) with a B-factor of $-100$ Å$^2$ and an applied dose-weighting scheme of 0.95 electrons/Å$^2$/frame. Omission of low-quality micrographs left a total of 1724 micrographs for downstream data processing. ~567,000 particles were picked in a reference-free manner using DoG picker (*Voss et al., 2009*). Global defocus and astigmatism were estimated using GCTF (*Zhang, 2016*) on the non-dose weighted aligned sums. Dose-weighted particles were binned to a final pixel size of 5.44 Å for an initial round of 2D classification using RELION/2.1 (*Kimanius et al., 2016*). 200,000 selected particles were re-centered, re-extracted, and unbinned to a final pixel size of 1.36 Å and subjected to 3D classification with RELION/2.1 (*Kimanius et al., 2016*) using the 30 Å low-pass filtered initial model generated from the DS-SOSIP D3$^\dagger$-VRC01$_{GL}$ negative-stain dataset. Out of the eight resulting classes, five classes contained well defined secondary structure elements and three bound Fabs. These classes were low-pass filtered to 20 Å and the best-resolved class was used as an initial model during 3D refinement using C3 symmetry. Refined angles for all particles were subsequently imported into FREALIGN (*Grigorieff, 2016*; *Lyumkis et al., 2013a*) and further refined with an applied particle weighting scheme. An additional iteration of refinement was performed by adjusting only the X/Y shifts. This refinement scheme resulted in a final estimated resolution of 3.8 Å for the three-Fab complex. The VRC01$_{GL}$ constant domains of the Fab were masked out during the final rounds of refinement and omitted from the final model due to the inherent flexibility of the elbow region (*Stanfield et al., 2006*). This same strategy was used for 3D classes of DS-SOSIP D3$^\dagger$-VRC01$_{GL}$ containing only two Fabs (C1 symmetry), leading to a final estimated resolution of 4.8 Å. Reported resolutions are based on the gold-standard FSC = 0.143 criterion. Local resolution estimates were generated using the ResMap software (*Kucukelbir et al., 2014*).

## Model building and refinement

We selected a clade A HIV-1 BG505 SOSIP.664 trimer (*Stewart-Jones et al., 2016*) and the 426c. TM1deltaV1-V3gp120 in complex with germline NIH46-46 (*Scharf et al., 2016*) as initial reference models for building 426c DS-SOSIP D3$^\dagger$-VRC01$_{GL}$. This model was manually trimmed and edited using Coot (*Emsley and Cowtan, 2004*; *Emsley et al., 2010*) and RosettaES (*Frenz et al., 2017*). We then further refined the structure in Rosetta using density-guided protocols (*Wang et al., 2016*) for the 3.8 Å resolution C3 reconstruction. This process was repeated iteratively until convergence and high agreement with the map was achieved. The Fab constant domains were masked out during refinement and omitted from the final model. Following refinement of protein coordinates, identified *N*-linked glycans were manually docked into their corresponding density and refined using Rosetta (*DiMaio et al., 2011*; *Frenz et al., 2018*). Multiple rounds of minimization were performed on the complete glycoprotein model and manually inspected for errors. Throughout this process, we applied strict non-crystallographic symmetry constraints in Rosetta (*DiMaio et al., 2011*). The 4.8 Å asymmetric 426c DS-SOSIP D3$^\dagger$-VRC01$_{GL}$ structure bound to only two Fabs was obtained by removing one of the Fabs bound to the aforementioned model and was rigid-body docked into the 2 Fab-bound map using UCSF Chimera.. Mannose rings not supported by density in this map were

manually trimmed. Final model quality was analyzed using Molprobity (*Chen et al., 2010*) and EM ringer (*Barad et al., 2015*). All figures were generated with UCSF Chimera (*Pettersen et al., 2004*).

## VRC01$_{GL}$-based Immunoprecipitation

Purified recombinant VRC01$_{GL}$ IgG was covalently coupled to Dnyabeads MyOne Tosylactivated beads (Life Technologies), and immunoprecipitation using magnetic separation was carried out according to the manufacturer's protocol. 5 mg of 426c core produced using HEK293S GnTI$^{-/-}$ cells were incubated with 100 µg of VRC01$_{GL}$-beads for 15 min (first bind), after which the beads were removed and unbound material (flow through) was incubated with a second fresh 100 µg aliquot of VRC01$_{GL}$-beads for an additional 15 min (second bind). VRC01$_{GL}$-beads from first and second binding were washed 3x before acidic elution and pH neutralization of affinity-purified samples. Unbound material was further depleted by incubation with a third 100 µg of VRC01$_{GL}$-beads, which were removed, before analysis. Samples of the original input 426c core, and VRC01$_{GL}$-bound and unbound fractions were resolved by SDS gel electrophoresis under reducing conditions and the remainder subjected to LC-MS/MS analysis, as described above.

## Mass spectrometry

For analysis of *N*-linked glycosylation profiles, an estimated 250 pmol of each HIV-1 426c-based construct analyzed in this paper was denatured, reduced, and alkylated by dilution to 5 µM in 50 µL of buffer containing 100 mM Tris (pH 8.5), 10 mM Tris(2-carboxyethylphosphine (TCEP), 40 mM iodoacetamide or 40 mM iodoacetic acid, and 2% (wt/vol) sodium deoxycholate. Samples were first heated to 95°C for 10 min and then incubated for an additional 30 min at room temperature in the dark. The samples were digest with trypsin (Sigma Aldrich), by diluting 20 µL of sample to total volume of 100 µL 50 mM ammonium bicarbonate (pH 8.5). Protease was added to the samples in a ratio of 1:75 by weight and left to incubate at 37°C overnight. After digestion, 2 µL of formic acid was added to the samples to precipitate the sodium deoxycholate from the solution. After centrifugation at 17,000 × g for 25 min, 85 µL of the supernatant was collected and centrifuged again at 17,000 × g for 5 min to ensure removal of any residual precipitated deoxycholate. 80 µL of this supernatant was collected. For each sample, 8 µL was injected on a Thermo Scientific Orbitrap Fusion Tribrid mass spectrometer. A 35 cm analytical column and a 3 cm trap column filled with ReproSil-Pur C18AQ 5 µM (Dr. Maisch) beads were used. Nanospray LC-MS/MS was used to separate peptides over a 90 min gradient from 5% to 30% acetonitrile with 0.1% formic acid. A positive spray voltage of 2100 was used with an ion transfer tube temperature of 350°C. An electron-transfer/higher-energy collision dissociation ion-fragmentation scheme (*Frese et al., 2013*) was used with calibrated charge-dependent entity-type definition (ETD) parameters and supplemental higher-energy collision dissociation energy of 0.15. A resolution setting of 120,000 with an AGC target of $2 \times 10^5$ was used for MS1, and a resolution setting of 30,000 with an AGC target of $1 \times 10^5$ was used for MS2. Data were searched with the Protein Metrics Byonic software (*Bern et al., 2012*), using a small custom database of recombinant protein sequences including the proteases used to prepare the glycopeptides. Reverse decoy sequences were also included in the search. Specificity of the search was set to C-terminal cleavage at R/K (trypsin), allowing up to two missed cleavages, with EthcD fragmentation (b/y- and c/z-type ions). We used a precursor mass and product mass tolerance of 12 ppm and 24 ppm, respectively. Carbamidomethylation of cysteines was set as fixed modification, carbamidomethylation of the lysines and N-terminal amines were set as variable modifications, methionine oxidation as variable modification, pyroglutamate identification was set for both N-terminal glutamines and glutamates as a variable modification, and a concatenated N-linked glycan database (derived from the four software-included databases) was used to identify glycopeptides. All analyzed glycopeptide hits were manually inspected to ensure for quality and accuracy. Semi-quantitative LC-MS/MS of VRC01-based immunoprecipitation experiments were performed using Skyline (*MacLean et al., 2010*) with peak integration and LC-MS/MS searches imported from Byonic. Missed cleavages and post-translational modifications listed above for qualitative LC-MS/MS searches were included in the quantification of glycopeptides. All MS1 peak areas used for integration were manually inspected to ensure for quality and accuracy. Unbound fractions from two experimental replicates were pooled and injected as two technical replicates, whereas each 'bound'

fraction (first bind and second bind) were performed as two experimental and two technical replicates each.

## Acknowledgments

Research reported in this publication was supported by grants from the National Institute of General Medical Sciences under Award Number R01GM120553 (DV) and T32GM008268 (AJB), the National Institute of Allergy and Infectious Diseases under award numbers R01AI081625 (LS), R01AI104384 (LS), P01AI094419 (LS) and U19AI109632 (LS), a Pew Biomedical Scholars Award (DV), an Investigators in the Pathogenesis of Infectious Disease Award from the Burroughs Wellcome Fund (DV) the Netherlands Organization for Scientific Research (NWO, Rubicon 019.2015.2.310.006; JS) and the European Molecular Biology Organisation (EMBO, ALTF 933–2015; JS). The authors acknowledge the use of instruments at the Electron Imaging Center for NanoMachines supported by NIH (1U24GM116792, 1S10RR23057 and 1S10OD018111), NSF (DBI-1338135) and CNSI at UCLA. This work was also partly supported by the University of Washington Arnold and Mabel Beckman cryoEM center and the Proteomics Resource (UWPR95794). We also thank Priska von Haller for her assistance in using the mass spectrometry instruments. Support for this work was provided in part by the Intramural research program of the Vaccine Research Center, NIAID, NIH. We thank the J B Pendleton Charitable Trust for its generous support of Formulatrix robotic instruments. X-ray diffraction data was collected at the Berkeley Center for Structural Biology beamline 5.0.2, which is supported in part by the National Institute of General Medical Sciences. The Advanced Light Source is supported by the Director, Office of Science, Office of Basic Energy Sciences, of the United States Department of Energy under contract number DE-AC02-05CH11231. We thank members of the Structural Biology Section and Structural Bioinformatics Core, Vaccine Research Center, for discussions or comments on the manuscript and in particular Hui Geng for providing Env trimer used in crystallization trials and Reda Rawi for providing glycan statistics using the HIV database.

## Additional information

### Funding

| Funder | Grant reference number | Author |
| --- | --- | --- |
| National Institute of General Medical Sciences | T32GM008268 | Andrew J Borst |
| Nederlandse Organisatie voor Wetenschappelijk Onderzoek | Rubicon 019.2015.2.310.006 | Joost Snijder |
| European Molecular Biology Organization | ALTF 933–2015 | Joost Snijder |
| National Institute of Allergy and Infectious Diseases | R01 AI081625 | Leonidas Stamatatos |
| National Institute of Allergy and Infectious Diseases | R01 AI104384 | Leonidas Stamatatos |
| National Institute of Allergy and Infectious Diseases | P01 AI094419 | Leonidas Stamatatos |
| National Institute of Allergy and Infectious Diseases | U19 AI109632 | Leonidas Stamatatos |
| National Institute of General Medical Sciences | R01 GM120553 | David Veesler |
| Pew Charitable Trusts | Biomedical Scholars Award | David Veesler |
| Burroughs Wellcome Fund | Investigators in the Pathogenesis of Infectious Disease Award | David Veesler |

The funders had no role in study design, data collection and interpretation, or the decision to submit the work for publication.

## Author contributions

Andrew J Borst, Conceptualization, Formal analysis, Validation, Investigation, Methodology, Writing—original draft, Writing—review and editing; Connor E Weidle, Conceptualization, Formal analysis, Investigation; Matthew D Gray, Conceptualization, Formal analysis, Validation, Investigation, Methodology; Brandon Frenz, Software, Validation, Methodology; Joost Snijder, Conceptualization, Formal analysis, Methodology, Writing—review and editing; M Gordon Joyce, Ivelin S Georgiev, Guillaume BE Stewart-Jones, Resources; Peter D Kwong, Resources, Writing—review and editing; Andrew T McGuire, Conceptualization, Formal analysis, Investigation, Writing—review and editing; Frank DiMaio, Software, Supervision, Validation, Methodology; Leonidas Stamatatos, Conceptualization, Resources, Formal analysis, Supervision, Funding acquisition, Validation, Project administration, Writing—review and editing; Marie Pancera, Conceptualization, Formal analysis, Supervision, Validation, Investigation, Project administration, Writing—review and editing; David Veesler, Conceptualization, Resources, Formal analysis, Supervision, Funding acquisition, Validation, Investigation, Methodology, Writing—original draft, Project administration, Writing—review and editing

## Author ORCIDs

Andrew J Borst (iD) https://orcid.org/0000-0003-4297-7824
Frank DiMaio (iD) http://orcid.org/0000-0002-7524-8938
David Veesler (iD) http://orcid.org/0000-0002-6019-8675

## Decision letter and Author response

Decision letter https://doi.org/10.7554/eLife.37688.041
Author response https://doi.org/10.7554/eLife.37688.042

# Additional files

**Supplementary files**

• Supplementary file 1. BLI kinetics parameters of various 426c core constructs expressed using HEK293F cells.
DOI: https://doi.org/10.7554/eLife.37688.020

• Supplementary file 2. RMSD Table comparing unliganded and liganded $VRC01_{GL}$ CDRL1 loop conformations.
DOI: https://doi.org/10.7554/eLife.37688.021

• Supplementary file 3. BLI kinetics parameters of various 426c core constructs expressed using $GnTI^{-/-}$ cells.
DOI: https://doi.org/10.7554/eLife.37688.022

• Transparent reporting form
DOI: https://doi.org/10.7554/eLife.37688.023

## Data availability

Mass spectrometry data have been deposited to the PRIDE archive under accession number PXD011494. CryoEM maps are available for download from the EMDB under accession numbers EMD-9294 ($426cDS-SOSIP\ D3^{\dagger}-VRC01_{GL}$, 3 Fabs, sharpened), EMD-9295 ($426cDS-SOSIP\ D3^{\dagger}-VRC01_{GL}$, 3 Fabs, unsharpened), EMD-9304 ($426cDS-SOSIP\ D3^{\dagger}-VRC01_{GL}$, 2 fabs, unsharpened), and EMD-9303 ($426cDS-SOSIP\ D3^{\dagger}-VRC01_{GL}$, 2 fabs, sharpened). Structures have been deposited to the PDB under accession numbers PDB 6MYY ($426cDS-SOSIP\ D3^{\dagger}-VRC01_{GL}$, 3 Fabs]), PDB-6MZJ ($426cDS-SOSIP\ D3^{\dagger}-VRC01_{GL}$, 2 fabs), and 6MFT ($426c\ core^{\dagger}-VRC01_{GL}$)

The following datasets were generated:

| Author(s) | Year | Dataset title | Dataset URL | Database and Identifier |
|---|---|---|---|---|
| Andrew J Borst, Connor E Weidle, Matthew D Gray, Brandon Frenz, | 2018 | Germline VRC01 antibody recognition of a modified clade C HIV-1 envelope trimer, 3 Fabs bound, sharpened map | http://www.ebi.ac.uk/pdbe/entry/emdb/EMD-9294 | EMBL-EBI Protein Data Bank, EMD-9294 |

| | | | | |
|---|---|---|---|---|
| Joost Snijder, M Gordon Joyce, Ivelin S Georgiev, Guillaume BE Stewart-Jones, Peter D Kwong, Andrew T McGuire, Frank DiMaio, Leonidas Stamatatos, Marie Pancera, David Veesler | | | | |
| Andrew J Borst, Connor E Weidle, Matthew D Gray, Brandon Frenz, Joost Snijder, M Gordon Joyce, Ivelin S Georgiev, Guillaume BE Stewart-Jones, Peter D Kwong, Andrew T McGuire, Frank DiMaio, Leonidas Stamatatos, Marie Pancera, David Veesler | 2018 | Germline VRC01 antibody recognition of a modified clade C HIV-1 envelope trimer, 2 Fabs bound, unsharpened map | http://www.ebi.ac.uk/pdbe/entry/emdb/EMD-9304 | EMBL-EBI Protein Data Bank, EMD-9304 |
| Andrew J Borst, Connor E Weidle, Matthew D Gray, Brandon Frenz, Joost Snijder, M Gordon Joyce, Ivelin S Georgiev, Guillaume BE Stewart-Jones, Peter D Kwong, Andrew T McGuire, Frank DiMaio, Leonidas Stamatatos, Marie Pancera, David Veesler | 2018 | Germline VRC01 antibody recognition of a modified clade C HIV-1 envelope trimer, 2 Fabs bound, sharpened map | https://www.rcsb.org/structure/6MZJ | RSCB Protein Data Bank, 6MZJ |
| Andrew J Borst, Connor E Weidle, Matthew D Gray, Brandon Frenz, Joost Snijder, M Gordon Joyce, Ivelin S Georgiev, Guillaume BE Stewart-Jones, Peter D Kwong, Andrew T McGuire, Frank DiMaio, Leonidas Stamatatos, Marie Pancera, David Veesler | 2018 | Germline VRC01 antibody recognition of a modified clade C HIV-1 envelope trimer, 2 Fabs bound, sharpened map | http://www.ebi.ac.uk/pdbe/entry/emdb/EMD-9303 | EMBL-EBI Protein Data Bank, EMD-9303 |
| Andrew J. Borst, Connor E. Weidle, Matthew D. Gray, Brandon Frenz, Joost Snijder, M. Gordon Joyce, Ivelin S. Georgiev, Guillaume B.E. Stewart-Jones, Peter D. Kwong, Andrew T. McGuire, Frank DiMaio, Leonidas Stamatatos, Marie Pancera, David Veesler | 2018 | Germline VRC01 antibody recognition of a modified clade C HIV-1 envelope trimer, 3 Fabs bound, sharpened map | https://www.rcsb.org/structure/6MYY | RCSB Protein Data Bank, 6MYY |

| Andrew J Borst, Connor E Weidle, Matthew D Gray, Brandon Frenz, Joost Snijder, M Gordon Joyce, Ivelin S Georgiev, Guillaume BE Stewart-Jones, Peter D Kwong, Andrew T McGuire, Frank DiMaio, Leonidas Stamatatos, Marie Pancera, David Veesler | 2018 | Germline VRC01 antibody recognition of a modified clade C HIV-1 envelope trimer, 3 Fabs bound, unsharpened map | http://www.ebi.ac.uk/pdbe/entry/emdb/EMD-9295 | EMBL-EBI Protein Data Bank, EMD-9294 |
|---|---|---|---|---|
| Andrew J. Borst, Connor E. Weidle, Matthew D. Gray, Brandon Frenz, Joost Snijder, M. Gordon Joyce, Ivelin S. Georgiev, Guillaume B.E. Stewart-Jones, Peter D. Kwong, Andrew T. McGuire, Frank DiMaio, Leonidas Stamatatos, Marie Pancera, David Veesler | 2018 | Recombinant HIV-1 426c Env Tryptic Digest Glycan Identifications and Semi-Quantification | https://www.ebi.ac.uk/pride/archive/projects/PXD011494 | EBI PRIDE, PXD011494 |
| Andrew J Borst, Connor E Weidle, Matthew D Gray, Brandon Frenz, Joost Snijder, M Gordon Joyce, Ivelin S Georgiev, Guillaume BE Stewart-Jones, Peter D Kwong, Andrew T McGuire, Frank DiMaio, Leonidas Stamatatos, Marie Pancera, David Veesler | 2018 | Crystal structure of glycosylated 426c HIV-1 gp120 core G459C in complex with glVRC01 A60C heavy chain | https://www.rcsb.org/structure/6MFT | RCSB Protein Data Bank, 6MFT |

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
