## [Decision Letter]

Thank you for submitting your article "Structural basis for germline VRC01 antibody recognition of a glycosylated HIV-1 envelope CD4-binding site" for consideration by *eLife*. Your article has been reviewed by three peer reviewers, one of whom is a member of our Board of Reviewing Editors, and the evaluation has been overseen by Tadatsugu Taniguchi as the Senior Editor. The reviewers have opted to remain anonymous.

The reviewers have discussed the reviews with one another and the Reviewing Editor has drafted this decision to help you prepare a revised submission. You will see that the reviewers find your results intriguing and wish to encourage the return of a revised submission, however, there appear to be a number of issues to be fully addressed, and that at least some further characterization will be required for this work to be reconsidered for publication in *eLife*.

The manuscript "Structural basis for germline VRC01 antibody recognition of a glycosylated HIV-1 envelope CD4-binding site" by Borst et al. attempts to structurally and biochemically describe the interaction of VRC01 with variants of the 426c Env containing the N276 glycan. If true, this would be the first time that a "wildtype" Env was shown to bind the inferred germline (iGL) of a broadly neutralizing antibody (bnAb) targeting the CD4-binding site (CD4_BS_). The main conclusion of this study, that VRC01-class inferred germlines (VRC01_GL_) can interact with Envs having short N276 glycans, is important because using Envs with short glycan chains in vaccination protocols may be preferred over deleting the N276 glycan to better guide the development of VRC01-class bnAbs. However, these results are quite surprising considering that some of the same authors published a manuscript in 2013 (McGuire et al., 2013 – also cited in this manuscript) demonstrating no observable binding of gp120s with an intact N276 glycan site to the iGL of VRC01 (the same CD4_BS_ bnAb used in this manuscript). Although we agree with the authors' conclusion that it might be best to use an immunogen containing CD4_BS_ glycans, there are several major issues described below that do not support the central theme – that VRC01_GL_s can bind to an Env with an N276 glycan. Most importantly, we believe that the authors must do definitive experiments to show that the N276 glycan can be engaged on a trimeric Env by the VRC01_GL_.

Essential revisions:

1) The authors are correct when they state "Our observation that a VRC01-class germline antibody bound to a CD4_BS_ in the presence of the Asn276 NLGS and its associated glycan (Figure 4D) is unprecedented". Although they reference the 426c gp120 core structure in this sentence, the perplexing result is the BioLayer Interferometry (BLI) data described in Figure 4A, which is virtually identical to an experiment the same authors published in 2013 (McGuire et al., 2013), but with the opposite result. Unless the sequences of the VRC01_GL_ or 426c Env are different, the result should not change (unless the glycan is not present in their antigen). Since this result is central to their story, it is essential the authors address this point. How are we to interpret the difference between their previous result and the result in the present paper?

2) The only interactions in the crystal structure of VRC01_GL_-426c gp120 with the N276 glycan are with the core GlcNAcs, which are the only part of the 276 glycan that is ordered. Mass spec data showed a predominance of (GlcNAc)_2_-(Man)_5_ glycans at position 276 in gp120s that were disulfide-linked with the VRC01_GL_, but if there was an interaction with the mannose portion of the glycan, one would have expected additional ordered glycan residues beyond the two GlcNAcs in the crystal structure. We're concerned that the interaction between VRC01_GL_ and 426c gp120 was "forced" by the disulfide bond connecting the two proteins, so even the interaction with the GlcNAcs may not be real.

3) The cryo-EM structure is beautiful and it's nice to see a structure with a GL Ab bound to a SOSIP Env, but unfortunately, the 276 glycan was removed in the SOSIP used for the cryo-EM structure. Thus, although described in the title of this manuscript, the N276 glycan and its interactions with VRC01_GL_ are not present in their cryo-EM structure and present in only one of the two molecules in the asymmetric unit of their crystal structure (and there only two GlcNAc residues are ordered). To obtain these structures the authors introduced a disulfide bond to stabilize the interaction between VRC01_GL_ and 426c gp120. Due to the artificial nature of these structures, the lack of extensive interactions with and glycan density for N276, and the fact that the VRC01_GL_ interactions with gp120 in the cryoEM and crystal structures have already been described (e.g., in VRC01_GL_ complexes with eODs), the title and Abstract is misleading to the reader: the manuscript never truly describes the structural basis of the N276 glycan interactions because the only interactions are with the core GlcNAcs – there are no other interactions, and there is no explanation for a preference for short mannose glycoforms at N276.

4) Although their binding studies demonstrate that removal of the N276 glycan increases affinity for VRC01_GL_, the authors also assert that VRC01_GL_ binds to a "wildtype" 426c gp120 and to 426c DS-SOSIP, both containing the N276 glycan. However, the binding they see could result from interactions with a subset of monomer or trimer that is not glycosylated, and not due to other reasons they mention (a Thr at position 278 would most likely not cause a steric clash since eOD-GT6 and eOD-GT8 have an Arg at this position and forms a complex with VRC01_GL_ in PDB ID 4JPK). Importantly, if this binding is due to a subset of gp120 (or gp140) that lacks a glycan at N276 (as shown in their crystal structure in which one of the molecules in the asymmetric unit apparently had an unglycosylated N276), then there is little experimental basis for the conclusion that VRC01_GL_ can bind to gp120s with a glycosylated N276 (also taking into account the artificial covalent linkage in the structures that force the interaction). Thus, the last sentence of the Abstract, "Our results show that engagement of the VRC01 germline antibody by a wildtype 426c gp120 is possible…", is not rigidly proven by the data presented in this paper; hence our request that the authors show that the N276 glycan can be engaged on a trimeric Env by the VRC01_GL_.

5) Figure 4D (middle panel) shows the interaction between VRC01 (mature) and the 276 glycan on JR-FL with only two GlcNAcs of that glycan shown – however, in the structure of this complex (Stewart-Jones et al., 2016), they resolved a (GlcNAc)_2_-(Man)_6_ glycan at this position, and they saw additional interactions with mannoses. The point of Figure 4D is to say that VRC01_GL_ interacts with the 276 glycan in a similar way as mature VRC01, but the authors don't show the full interaction that was seen in the VRC01 (mature) – Env structure.

6) The affinities reported in Figures 1B, 2A-D and 5B-E are derived from only binding experiments involving only four or five IgG concentrations. This might be alright (although more concentrations would be preferable). However, an accurate affinity can only be derived if the experiment includes data that is 10-fold above and 10-fold below the concentration corresponding to the K_D_. Note the K_D_ reported in Figure 1B is not accurate because of avidity effects, which are mentioned by the authors, but they should not report a K_D_ for this type of experiment.

7) What is the binding affinity of VRC01_GL_-A60C_HC_ to 426c G459C DS-SOSIP HIV-1 SOSIP trimers?

8) Regarding the cryo-EM structure: (a) the authors should show map-to-model FSCs to validate the model. (b) Please also show a local resolution analysis and Euler distribution plot. (c) Table 1 should also indicate additional information: number of micrographs, dose rate, cross-correlation of fitted model to map (to complement Molprobity and Clash scores). We have found that EM-Ringer is also a great tool to assess the quality of the modeled side-chains. Even though some of this is redundant with what is written in the methods, it's easier for the reader to have all information in one place. (d) In Figure 7, please specify how many particles were used for the 3D reconstruction. The structure of Env should be docked into the density to better judge the reconstructions.

9) Figure 1. It would be helpful to show comparisons of iGL and mature VRC01 interactions with gp120.

---

## [Author Response]

Essential revisions:1) The authors are correct when they state "Our observation that a VRC01-class germline antibody bound to a CD4_BS_ in the presence of the Asn276 NLGS and its associated glycan (Figure 4D) is unprecedented". Although they reference the 426c gp120 core structure in this sentence, the perplexing result is the biolayer interferometry (BLI) data described in Figure 4A, which is virtually identical to an experiment the same authors published in 2013 (McGuire et al., 2013), but with the opposite result. Unless the sequences of the VRC01_GL_ or 426c Env are different, the result should not change (unless the glycan is not present in their antigen). Since this result is central to their story, it is essential the authors address this point. How are we to interpret the difference between their previous result and the result in the present paper?

We would like to emphasize that nothing reported in this manuscript contradicts prior work by our groups or others regarding the 426c strain of HIV-1. In the McGuire et al. (2013) reference, we examined the binding of germline antibodies to soluble WT 426c gp140s or 426c gp140 lacking NLGS in Loop D (N276) and V5 (N460 and N463). However, in this manuscript, we are performing experiments using the 426c core. This construct lacks the variable loops 1, 2 and 3 and gp41, but retains the aforementioned 3 NLGSs.

We point to McGuire et al. (2016) and McGuire et al. (2014), where we demonstrated that removal of these variable loops from a version of the 426c core that already lacked the above mentioned 3 NLGSs significantly increased binding of VRC01_GL_ antibodies to the CD4_BS_. This manuscript follows-up on the observations made in McGuire et al. (2016) to now include that VRC01_GL_ can engage variants of these variable region-deleted 426c core constructs with wild-type NLGSs. This experiment is completely novel. As a result, the data reported here do not contradict any prior work performed by us, or others, regarding the 426c strain.

We would also like to note that stabilized trimeric pre-fusion closed SOSIP constructs likely impart additional negative steric effects, not present in monomeric gp120 constructs, due to the presence of neighboring protomers and additional glycans protruding towards the CD4_BS_ from these protomers (i.e. Asn262 and others). We have edited the corresponding sections of the manuscript and remade select figures to be clearer in this regard. We also included more references to the two aforementioned pieces of literature and more clearly highlighted the differences between our previously published work, the significance of the findings in this current manuscript, and how we are building upon prior characterization performed by our groups.

2) The only interactions in the crystal structure of VRC01_GL_-426c gp120 with the N276 glycan are with the core GlcNAcs, which are the only part of the 276 glycan that is ordered. Mass spec data showed a predominance of (GlcNAc)_2_-(Man)_5_ glycans at position 276 in gp120s that were disulfide-linked with the VRC01_GL_, but if there was an interaction with the mannose portion of the glycan, one would have expected additional ordered glycan residues beyond the two GlcNAcs in the crystal structure.

We agree that the only glycan moieties that appear to interact with VRC01_GL_ in the crystal structure are the two proximal GlcNAcs. We do not suspect the mannose rings are involved in the binding directly, but rather act as steric barrier that VRC01_GL_ must overcome in order to engage the CD4_BS_. This is made evident by the difference in observed binding affinities between 426c gp120c constructs expressed in GnTI^-/-^ cells in either the presence or absence of kifunensine (which enriches for (GlcNAc)_2_-(Man)_9_ glycans when kifunensine is present at 100 µM during expression). We also did not detect the presence of (GlcNAc)_2_ moieties in this complex by LC-MS/MS, suggesting the mannose rings in the crystal structure are present in the complex, but are disordered. Overall, we agree with the reviewers on their interpretation of the crystal structure and have made this clearer in the revised manuscript and its associated figures.

We're concerned that the interaction between VRC01_GL_ and 426c gp120 was "forced" by the disulfide bond connecting the two proteins, so even the interaction with the GlcNAcs may not be real.

Although we appreciate the reviewers’ concern, it is unlikely that the engineered disulfide would force the VRC01_GL_ and 426 gp120 to interact in a homogeneous way yielding diffraction-quality crystals. The engineered disulfide will only prevent the dissociation of the two proteins, similarly to what we previously described for mature VRC01 (Stewart-Jones et al., 2016), but will not force them to ‘come together’ in the first place.

Moreover, we have performed a VRC01_GL_-basedimmunoprecipitation experiment on the 426c gp120 core and subjected both the bound and unbound fractions to SDS-PAGE and LC-MS/MS for semi-quantitative glycoproteomics analysis. Our results showed that bound fractions had an increased electrophoretic mobility compared to the unbound fraction. LC-MS/MS revealed an enrichment for un-glycosylated Asn276 in the bound fraction relative to the unbound fraction, suggesting this subspecies is the preferred binder. However, we also detected a large fraction of (GlcNAc)_2_-(Man)_5_ glycan at position Asn276 in the bound fraction as well, strongly suggesting binding is still occurring in the presence of an Asn276 glycan (and in the absence of the disulfide cross-link). Other sites were also measured, including those on the V5 loop, revealing glycosylation profiles that replicate what was observed structurally. These results strongly support our interpretation of the cross-linked crystal structure showing both the glycosylated and un-glycosylated Asn276 glycoforms are present in solution and that both engage VRC01 _GL_ in the context of soluble monomeric 426c core gp120. This result does not disagree with the current dogma that an un-glycosylated glycoform is a preferred binder, but rather supports it while also expanding our understanding of VRC01_GL_ binding requirements to include monomeric core gp120 containing an Asn276 short glycan with a lower binding affinity.

3) The cryo-EM structure is beautiful and it's nice to see a structure with a GL Ab bound to a SOSIP Env, but unfortunately, the 276 glycan was removed in the SOSIP used for the cryo-EM structure. Thus, although described in the title of this manuscript, the N276 glycan and its interactions with VRC01_GL_ are not present in their cryo-EM structure and present in only one of the two molecules in the asymmetric unit of their crystal structure (and there only two GlcNAc residues are ordered). To obtain these structures the authors introduced a disulfide bond to stabilize the interaction between VRC01_GL_ and 426c gp120. Due to the artificial nature of these structures, the lack of extensive interactions with and glycan density for N276, and the fact that the VRC01_GL_ interactions with gp120 in the cryoEM and crystal structures have already been described (e.g., in VRC01_GL_ complexes with eODs), the title and Abstract is misleading to the reader: the manuscript never truly describes the structural basis of the N276 glycan interactions because the only interactions are with the core GlcNAcs – there are no other interactions, and there is no explanation for a preference for short mannose glycoforms at N276.

Although the title may have been overstated during our initial submission, we believe our inclusion of semi-quantitative LC-MS/MS data strongly supports binding of VRC01_GL_ to the glycosylated CD4_BS_ of the 426c core construct is indeed occurring in the absence of crosslinking (Figure 3I). Nonetheless, we have revised the manuscript title to be more conservative.

Our structure, endoglycosidase treatment assays, and semi-quantitative LC-MS/MS analyses on VRC01_GL_-based IP experiments strongly suggests that the interactions with core GlcNAcs are important elements of the VRC01_GL_ interaction. We believe describing these interactions provides a blueprint for understanding Asn276 glycan accommodation by VRC01_GL_. Direct interactions with more distal glycan rings might be restricted to mature VRC01 since the residues involved in these contacts are mutated in mature VRC01 compared to VRC01_GL_. We have included the corresponding data and explanations in the manuscript for rigor and clarity.

4) Although their binding studies demonstrate that removal of the N276 glycan increases affinity for VRC01_GL_, the authors also assert that VRC01_GL_ binds to a "wildtype" 426c gp120 and to 426c DS-SOSIP, both containing the N276 glycan. However, the binding they see could result from interactions with a subset of monomer or trimer that is not glycosylated, and not due to other reasons they mention (a Thr at position 278 would most likely not cause a steric clash since eOD-GT6 and eOD-GT8 have an Arg at this position and forms a complex with VRC01_GL_ in PDB ID 4JPK).

We agree that this was a possibility and have thus addressed this concern with the addition of aforementioned immunoprecipitation experiments and associated mass spectrometry analysis of bound and unbound fractions. Our data strongly supports our crystal structure suggesting both (GlcNAc)_2_-(Man)_5_ and un-glycosylated Asn276 glycoforms interact with VRC01_GL_ (data taken from the “IP-bound” fraction).

Regarding the comments about Threonine 278 likely not causing a steric clash: we understand the reviewers’ point and have decided to directly address it with new experiments.

We performed affinity measurements using the following 426c core constructs: S278, S278T, S278R, S278A, and S278V and observed a higher affinity for all mutants relative to the native S278 variant of the 426c core. This rules out that a bulky and/or branched residue at this position contributed to the difference in binding. We have included all of this data in the revised version of the manuscript (Figure 6—figure supplement 1).

Importantly, if this binding is due to a subset of gp120 (or gp140) that lacks a glycan at N276 (as shown in their crystal structure in which one of the molecules in the asymmetric unit apparently had an unglycosylated N276), then there is little experimental basis for the conclusion that VRC01_GL_ can bind to gp120s with a glycosylated N276 (also taking into account the artificial covalent linkage in the structures that force the interaction). Thus, the last sentence of the Abstract, "Our results show that engagement of the VRC01 germline antibody by a wildtype 426c gp120 is possible…", is not rigidly proven by the data presented in this paper; hence our request that the authors show that the N276 glycan can be engaged on a trimeric Env by the VRC01_GL_.

This concern was addressed with the inclusion of the IP experimental data in combination with the semi-quantitative LC-MS/MS showing the presence of an Asn276 glycan in the bound fraction of the 426c core construct. We believe both species are compatible with VRC01_GL_ binding, with the un-glycosylated variant being preferred. This statement is currently supported by our crystal structure, our reported BLI binding data and associated EndoH digestion experiment, increasing levels of binding as glycans become shorter, and the aforementioned IP pulldowns and LC-MS/MS which we have included in the revised version this manuscript (Figure 3I). Although we agree that a WT 426c DS-SOSIP structure would have been preferred, we do not believe that a structure of an Env trimer with a glycan at position Asn276 in complex with VRC01_GL_ would bring additional information regarding the utility of the 426c core for VRC01_GL_ antibody recognition, nor is it feasible due to the low affinity of this complex. We have included binding data of trimeric 426c DS-SOSIP and VRC01_GL_ in Figure 1A. We wish to emphasize in our revision that using a wild-type core 426c gp120 in the absence of variable loops V1, V2, and V3 permits binding to VRC01_GL_ that is otherwise only detected at very low levels using SOSIP trimers.

5) Figure 4D (middle panel) shows the interaction between VRC01 (mature) and the 276 glycan on JR-FL with only two GlcNAcs of that glycan shown – however, in the structure of this complex (Stewart-Jones et al., 2016), they resolved a (GlcNAc)_2_-(Man)_6_ glycan at this position, and they saw additional interactions with mannoses.

We apologize for this oversight and have re-rendered the figure to include the additional interactions with the mannose moieties in mature VRC01.

The point of Figure 4D is to say that VRC01_GL_ interacts with the 276 glycan in a similar way as mature VRC01, but the authors don't show the full interaction that was seen in the VRC01 (mature) – Env structure.

The point of this figure was to show differences between the two structures, not similarities. We have made this clearer in the manuscript and figure legend. We have also included the full structure of the Asn276 glycan in the mature VRC01 structure (Figure 3D).

6) The affinities reported in Figures 1B, 2A-D, and 5B-E are derived from only binding experiments involving only four or five IgG concentrations. This might be alright (although more concentrations would be preferable). However, an accurate affinity can only be derived if the experiment includes data that is 10-fold above and 10-fold below the concentration corresponding to the K_D_. Note the K_D_ reported in Figure 1B is not accurate because of avidity effects, which are mentioned by the authors, but they should not report a K_D_ for this type of experiment.

We agree with the reviewer’s comments and have redone the experiments using a more appropriate concentration range. For the SOSIP trimer, we have reported the affinity measurement as “apparent affinity” to account for avidity effects and made this clear in the main text and in Figure 1B.

7) What is the binding affinity of VRC01_GL_-A60C_HC_ to 426c G459C DS-SOSIP HIV-1 SOSIP trimers?

The binding affinity is very low, suggesting stabilized WT SOSIP constructs are poor binders to VRC01_GL_-class antibodies, as we reported previously. 426c gp120 core constructs appear to bind much better with a wild-type sequence lacking V1, V2, and V3 loops relative to the SOSIP counterpart. We have emphasized this in the resubmission and include BLI measurements of the VRC01_GL_ to WT 426c DS-SOSIP trimer in Figure 1B.

8) Regarding the cryo-EM structure: (a) the authors should show map-to-model FSCs to validate the model. (b) Please also show a local resolution analysis and Euler distribution plot. (c) Table 1 should also indicate additional information: number of micrographs, dose rate, cross-correlation of fitted model to map (to complement Molprobity and Clash scores). We have found that EM-Ringer is also a great tool to assess the quality of the modeled side-chains. Even though some of this is redundant with what is written in the methods, it's easier for the reader to have all information in one place. (d) In Figure 7, please specify how many particles were used for the 3D reconstruction. The structure of Env should be docked into the density to better judge the reconstructions.

We have included an updated cryoEM table with the requested information as well as a map-to-model FSC curve as part of the supplementary information (Figure 1—figure supplements 3-4). In an attempt to streamline the manuscript and better emphasize in our revision that the 426c core is a better binder than the SOSIP trimer, we have removed the original version Figure 7 and its associated section from the manuscript.

9) Figure 1. It would be helpful to show comparisons of iGL and mature VRC01 interactions with gp120.

We have included gp120 residues that form contacts with VRC01_GL_ and VRC01_MAT_ as Figure 1—figure supplement 5. We have also added a comparison between VRC01_GL_-class recognition differences between the core and trimeric 426c constructs in Figure 1G with an emphasis on the different conformation of the gp120 bridging sheet.